# Recent Achievements in Electrochemical and Optical Nucleic Acids Based Detection of Metal Ions

**DOI:** 10.3390/molecules27217481

**Published:** 2022-11-02

**Authors:** Marta Jarczewska, Anna Szymczyk, Joanna Zajda, Marcin Olszewski, Robert Ziółkowski, Elżbieta Malinowska

**Affiliations:** 1Chair of Medical Biotechnology, Faculty of Chemistry, Warsaw University of Technology, Stanisława Noakowskiego 3, 00-664 Warsaw, Poland; 2Doctoral School, Warsaw University of Technology, Plac Politechniki 1, 00-661 Warsaw, Poland; 3Chair of Analytical Chemistry, Faculty of Chemistry, Warsaw University of Technology, Stanisława Noakowskiego 3, 00-664 Warsaw, Poland; 4Chair of Drug and Cosmetics Biotechnology, Faculty of Chemistry, Warsaw University ofTechnology, Koszykowa 75, 00-664 Warsaw, Poland; 5Centre for Advanced Materials and Technologies CEZAMAT, Warsaw University of Technology, Poleczki 19, 02-822 Warsaw, Poland

**Keywords:** aptamers, nucleic acids, metal ions, electrochemistry, optical detection methods, nanomaterials, microflow devices

## Abstract

Recently nucleic acids gained considerable attention as selective receptors of metal ions. This is because of the possibility of adjusting their sequences in new aptamers selection, as well as the convenience of elaborating new detection mechanisms. Such a flexibility allows for easy utilization of newly emerging nanomaterials for the development of detection devices. This, in turn, can significantly increase, e.g., analytical signal intensity, both optical and electrochemical, and the same can allow for obtaining exceptionally low detection limits and fast biosensor responses. All these properties, together with low power consumption, make nucleic acids biosensors perfect candidates as detection elements of fully automatic portable microfluidic devices. This review provides current progress in nucleic acids application in monitoring environmentally and clinically important metal ions in the electrochemical or optical manner. In addition, several examples of such biosensor applications in portable microfluidic devices are shown.

## 1. Introduction

Metal ions detection is an important issue from the environmental and medical point of view [1]. Heavy metals can be treated as non-degradable pollutants. They are constantly released to surface and ground waters (industry, mining, agriculture), from where they are introduced into living organisms through the food chain system of the earth. As they easily accumulate in animals or the human body, they often lead to severe systematic damage (skeletal, central nervous system, kidneys, liver, reproductive systems, etc.). Their presence in living organisms induces, among others, free radical formation, lipid peroxidation, DNA damage, depletion of antioxidants, etc. [2]. That is why their high concentration or long-term exposure may also have carcinogenic effects [3]. Because of the above, the heavy metal ions concentration determination may be necessary to evaluate the possible environmental pollution or, in the case of live beings, their exposure and resulting contamination. Moreover, other metals, such as potassium or sodium, are important indicators of human health [4]. That is why metal ions monitoring is an important issue in environmental protection, disease prevention, and general healthcare [1,5]. Nonetheless, fast and reliable determination of metal ions is still a challenging task. This is primarily because of the complex composition of most biological and environmental samples, which also contain high concentrations of interfering components. Therefore, the selective capturing of target metal ions from such complex matrices is a key step in their concentration determination. In recent years, the use of nucleic acids, especially aptamers, as the selective biological receptors in various biosensors and assays has attracted significant interest [6].

## 2. Aptamer—The Bioreceptor

The principal purpose of the biological recognition element of the biosensor is to provide superior analyte selectivity. Diverse biorecognition elements exist, each with unique characteristics, ranging from naturally occurring to synthetic constructs. A distinct class constitutes pseudo-natural aptamers utilized in aptasensors [7]. Aptamers are small single-stranded DNA or RNA (usually consisting of 20–100 bases) that, with high affinity and specificity, bind to various targets, including ions, small molecules, peptides, proteins, viruses, bacteria, and even whole cells [8,9,10,11,12]. The nucleic acid analog aptamers such as peptide nucleic acids (PNA) or locked nucleic acids (LNA), as well as peptide aptamers composed of amino acids building blocks, have also been developed; however, their application in ions detection is relatively scarce [13]. The key features of aptamers have been summarized in Table 1. The term aptamer was introduced by Andy Ellington and Jack Szostak and is derived from the Latin word “Aptus”, meaning “to fit”, and the Greek suffix “meros”, meaning “part” [14]. As the name implies, aptamers are designed to “fit” their targets. In the event of molecular recognition, the single-stranded oligonucleotides fold into a specific three-dimensional conformational structure, creating a stable complex with a target ligand. This process involves intermolecular forces such as hydrogen bonds, stacking of aromatic rings, van der Waals forces, and hydrophobic and electrostatic interactions.

Unlike antibodies, to which they are often compared, aptamers are generated chemically through an in vitro process called Systematic Evolution of Ligands by EXponential enrichment (SELEX) and do not require the use of animals. This concept was independently introduced by two research groups over 30 years ago [14,15] and patented [16,17]. Since then, over 25 different SELEX processes have been implemented that follow basic Gold and Tuerk’s approach [16]. The general procedure involves the generation of the random oligonucleotide library (10^12^–10^15^ sequences) and incubation with the target molecule. Then, in the partitioning step, the target-bound aptamers are separated from unbound ones. Bound sequences are recovered and amplified by PCR or RT-PCR. Afterward, thus enriched pool is used in the next round of selection with increased stringent selection parameters. The oligonucleotide sequences with good specificity and affinity can typically be obtained in 6 up to 20 cycle rounds. The final oligonucleotide sequences are identified by cloning and Sanger sequencing. As mentioned above, numerous modifications have been made to the traditional SELEX process within the last three decades to obtain more efficient SELEX procedures. Current approaches and recent advancements in the SELEX method have already been summarized, and interested readers are referred to the previously published excellent reviews [18,19,20,21,22].

From the point of view of the aptasensors’ design, the simple and cost-effective chemical generation of aptamers ensures their reproducibility and activity in various batches. They can also be easily modified with functional moieties to meet the particular needs of the detection mode. Moreover, they are more stable in harsh conditions than natural receptors such as antibodies or enzymes and are readily regenerated after denaturation.

## 3. Ion Recognition and Signal Readout Methods

It has been demonstrated that ions can specifically interact with bases of DNA or RNA to form stable complexes. The ability of cations to bind to single-stranded oligonucleotides and induce secondary structures can be associated with their ionic radius, coordination behavior, and hydration effects [23]. In particular, numerous reports on stabilizing the G—quadruplex structure of guanine (G)—rich oligonucleotides by K^+^, Na^+^, Ca^2+^, and Pb^2+^ have been reported [24,25,26]. The ability of Cd^2+^ to be captured by the thymine (T) and G-rich sequence was also demonstrated. This interaction was ascribed to the presence of the lone electron pairs in the outer shell of cadmium that could bind to the adjacent T or G bases through the coordination bonds between Cd^2+^ and the O or N of those bases [27]. Additionally, it has been shown that T–T and cytosine—cytosine (C–C) mismatches have the capability to recognize Hg^2+^ and Ag^+^ exclusively, forming T–Hg^2+^–T and C–Ag^+^–C complexes, respectively, by the imino protons exchange [28,29]. Nonetheless, aptamer recognition of ions is challenging due to their simple structure and single binding site. Moreover, the formation of an aptamer-ion complex does not occur with appreciable mass change and steric hindrance effect. This limits the application of aptasensors in ion detection and poses additional requirements for the signal transducer and detection mode selection. Nevertheless, aptamers remain attractive bioligands due to their inherent advantages. In general, the formation of the aptamer-target complex can be probed with a range of methods; however, due to the abovementioned factors, electrochemical and optical are preferred for ion determination.

In the case of electrochemical aptasensors, aptamers are immobilized on the electrode surface, and the formation of aptamer-ion complexes is determined by examining the electron transfer features of the redox probe. On the one hand, the aptamer can be labeled with a redox probe that, upon binding with a target, is either close to the electrode surface or at a further distance, which results in enhanced or decreased electron transfer. In this “signal on/off” approach, methylene blue (MB) and ferrocene (Fc) are frequently utilized as redox probes. On the other hand, in label-free electrochemical aptasensors, the redox probe is in the sample solution and interacts with the immobilized oligonucleotide layer. The changes in the receptor layer induced by the ion capture can be detected with the probes that either intercalate in DNA/RNA structure (e.g., MB, crystal violet) [30,31] or electrostatically interact with the phosphate backbone (e.g., Fe(CN)_6_^4−/3−^, RuHex, AQMS) [13]. The introduction of a redox-active species enables the application of several techniques, including voltammetry, electrochemical impedance spectroscopy (EIS) [32], or field-effect transistors (FET) [33,34]. All of these techniques are dedicated to the electron flow measurement, which is generated by different factors, and various factors influence its intensity. The change in the electron flow intensity results from metal ion binding to the biosensor’s receptor layer and constitutes the analyzed analytical signal. As for voltammetric or EIS techniques, the current change results from the change in the kinetics or thermodynamics of redox-active specie redox reaction after metal binding by receptor layer [32]. The mechanism of detection for FET is slightly different. In this case, the analytical setup comprises three indispensable parts named (S) source, (D) drain, and gate, placed between these elements. In the case of FET devices, the current flows from the source to drain through the gate. As the receptor layer is immobilized at the gate surface, any changes in its electrical properties (e.g., an increase in the concentration of positively charged metal ions) significantly influence the change in the gate conductivity. This, in turn, results in the current flow change between the source and the drain, and at the same time, the biosensor response [33,34].

In optical aptasensors, fluorimetry and colorimetry are the most widely exploited techniques. The choice of fluorescence detection is dictated by the ease of aptamers labeling with the various fluorescent dyes and the inherent capability of real-time detection. Here, different approaches can also be distinguished. One of them employs a double end-labeled aptamer with a fluorophore and a quencher (or FRET acceptor-donor pair). The target binding-induced conformational changes lead to decreased distance between the fluorophore and the quencher, which results in decreased fluorescence (or increased in the case of the FRET pair) [35]. In the case of fluorescent dye (reporter) and quencher pair, static quenching occurs when the two molecules are in physical contact. As a result, the absorption spectrum of a reporter molecule is distorted. The FRET is a through-space mechanism in which the energy from the donor is transferred to the acceptor without absorption or emission of light. The phenomenon is highly distance-dependent and occurs when the molecules are approximately 10–100 Å apart. The donor’s emission spectrum has to overlap with the absorption spectra of the acceptor, and they are not affected by the distance. In the second format, an aptamer and a complementary strand in a duplex structure are labeled with a fluorophore and a quencher. While the complementary strand is separated from the aptamer upon target binding, the fluorescence is restored [36]. Additionally, the fluorophore-labeled aptamer can be adsorbed on single-walled carbon nanotubes that efficiently quench the dye’s fluorescence. Here, the fluorescence is likewise increased upon binding with the target ion [37]. As a quencher, graphene oxide and AuNPs are also frequently utilized [38,39]. Colorimetric aptasensors mainly employ gold nanoparticles and rely on the aggregation of AuNPs upon target binding associated with the concomitant red-shift that can be probed even with the naked eye [40]. The main disadvantage of the fluorescent and colorimetric aptasensors, however, is the interference of the background signal of a complex matrix.

For obvious reasons, mass-dependent methods, which do not require labels, are practically not utilized to detect ions. However, very few reports describing quartz crystal microbalance (QCM) based aptasensors can be traced in the literature [41]. In the following chapters of this review, particular aptasensor designs aiming at their performance improvement and novel detection modes are discussed in more details.

## 4. Aptasensors for Metal Ions Detection

To date, various methods have been developed for metal ions detection. They can be divided into two main groups, (i) traditional detection methods: atomic absorption/emission spectrometry, atomic fluorescence spectrometry, inductively coupled plasma mass spectrometry, high-performance liquid chromatography, etc. [1]; (ii) (bio)sensors: e.g., with enzymes, antibodies, nucleic acids in the receptor layer and with electrochemical, piezoelectric, or optical detection mechanism [42]. In the case of the first mentioned methods group, it is possible to accurately determine the metal ions concentration with high selectivity; however, it is indispensable to use on-site available professional and the same expensive equipment together with sample pretreatment, which limits the application of such methods [1]. The second mentioned group, (bio)sensors, depends on specific recognition elements such as enzymes/substrates, antigens/antibodies and targets/aptamers or simply electrode surface/target [42]. However, because functional nucleic acids are easy to synthesize and modify and are insensitive to a wide range of environmental parameters, they are a powerful tool for constructing sensors for metal ions detection [6].

### 4.1. Potassium Ion

Potassium ions are important electrolytes that participate in various processes including cell membrane physiology, particularly in providing the resting membrane potential and also in the generation of action potentials in the nervous system and heart, as well as maintenance of muscular strength, enzyme activation and regulation of blood pressure. It should be noted that potassium ions are transported into cells using sodium—potassium adenosine triphosphatase (Na, K- ATP-ase), which ensures intracellular potassium ion concentration that is at least 30—times higher than extracellular K^+^ concentration. The abnormal potassium ions concentration can lead either to Hypokalemia (serum concentration below 3.6 mM) or Hyperkalemia (serum concentration above 5 mM) as well as can be an indication of other diseases such as heart, diabetes, and AIDS [4]. In the case of aptamer sensing layer for potassium detection, the guanine-rich strands are utilized. It is known that in the presence of K^+^ ions, a four-stranded helical conformation with arrays of G-quadrats that exist thanks to the Hoogsteen-type base pairing is formed and named G-quadruplexes [43] (Figure 1).

#### 4.1.1. Electrochemical Potassium Ion Aptasensors

Historically, the first aptasensor elaborated for potassium ions detection was prepared by Radi and O’Sullivan [24]. In their work, an electrochemical aptasensor was developed, where Guanine—rich aptamers were applied as receptors in the sensing layer. In the presence of potassium ions, a random coil structure was changed into a compact G-quadruplex structure. As a result, a ferrocene label attached to the 5′ end of the sequence was moved to close proximity of the electrode surface, which led to the enhancement of the ferrocene anodic current (Figure 1). For the potassium ions concentration below 1 mM, a 1:1 complex with an aptamer was formed; whereas, for higher concentrations, a 2:1 complex was present. The proposed biosensor exhibited a linear response range from 0.1 to 1 mM with a lower limit of detection of 0.015 mM. The aptasensor showed high selectivity over interfering ions such as sodium, calcium, and magnesium, and the sensing layer was regenerated using 0.1 M HCl. The formation of a dense, compact layer in the presence of potassium ions was also evidenced using the impedance technique with the application of a ferri/ferrocyanide redox indicator.

A similar approach with the application of a 3′ thiolated aptamer strand that additionally contained a 5′ end with a ferrocene label as a receptor layer was proposed by Chen et al. [44]. In such a system, the addition of potassium ions led to the formation of a G-quadruplex structure and a dense monolayer that resulted in the smaller proximity of the ferrocene label from the electrode surface. Consequently, with the increased potassium ion concentration, a higher current response was observed using the square-wave voltammetry technique. The proposed system enabled potassium ions detection in the range from 0.1 to 50 nM with a LOD of 0.1 nM. The interference studies showed that a small current signal was recorded in the presence of 50 nM Ca^2+^, Mg^2+^, Na^+^, Li^+^, Fe^2+^, Zn^2+^, and NH_4_^+^. Finally, the proposed aptasensor was used for potassium ion detection in diluted serum samples, and recoveries of 97.79%, 99.20%, 99.46% and 100.54% were obtained.

The same group elaborated on an aptasensor that used spectroscopy impedance as a detection technique [45] (Figure 2). For that purpose, the following sequence was applied: 5′-TTTGGTTGGTGTGGTTGGTTT-3′- (CH_2_)_6_–SH, and the electrochemical experiments were conducted in the presence of ferri/ferrocyanide redox couple. The addition of potassium cations caused the formation of a G-quadruplex structure distinguished by higher space charge density that caused higher repulsion of redox indicator than in the absence of the analyte—potassium ions. The impedimetric response was linear in the range from 0.1 nM to 1 mM with a LOD of 0.1 nM. The negligible signal change was observed in the presence of the interfering cations, including 0.1 mM Ca^2+^, Mg^2+^, Na^+^, Li^+^, Al^3+^, Zn^2+^, Cu^2+^, and Ni^2+^. Finally, potassium ion concentration in urine samples was analyzed. It is known that normal potassium ion concentration in a such sample varies from 25 to 125 mM. In four filtrated and 100-fold diluted samples, potassium was detected within the normal concentration range. Spiked urine samples were also analyzed, and the recovery was at the level of 91.16–100.70%.

Other examples with an external redox indicator refer, e.g., to hemin, which was added to the aptasensing system, where the sequence: 5′-GGGTAGGGCGGGTTGGGAAA-3′-(CH_2_)_6_-SH was applied for potassium detection [46]. According to the Authors, the formation of a G-quadruplex in the presence of K^+^ ions allowed for aptamer binding with hemin, hence, causing the increase in DPV current signal. The proposed system enabled potassium detection in the range from 0.1 nM to 0.1 mM with a LOD of 10 nM. The interfering ions Ca^2+^, Mg^2+^, and NH_4_^+^ did not cause a pronounced current response in comparison to potassium ions.

Jarczewska et al. investigated various types of external redox indicators, which could be used in the development of electrochemical aptasensor for K^+^ ions of the best analytical parameters [47]. In this approach, three aptamer sequences were applied as receptor layers that were previously shown to bind selectively with potassium ions, namely TBA, PS2M, and AG3 aptamers that differed in length and nucleotide content. The studies allowed for the choice of redox indicator that was methylene blue at 1 µM concentration. The lowest dissociation constant showing the highest affinity of potassium ions towards the aptamer strand was obtained for the TBA sequence and K_d_ of 1.13 µmol L^−1^. The proposed receptor layer enabled potassium ions detection in the range from 10^−8^ to 10^−5^ mol L^−1^ and a lower limit of detection of 2.31 × 10^−9^ mol L^−1^. The elaborated sensor also showed good selectivity towards potassium ions with the most apparent interference from magnesium and lithium ions.

There are also examples where the gold electrode was additionally modified with different polymers or nanomaterials in order to obtain, e.g., lower detection limits. A good example of such is a biosensor, where the sensing layer was elaborated by the formation of the layer composed of p-aminothiophenol on the gold electrode surface. This was proceeded by the deposition of gold nanoparticles, which enhanced signal intensity and was an anchor point for the aptamer strand—5′-TTTGGTTGGTGTGGTTGGTTT- 3′-(CH_2_)_6_-SH. As a result, the small intensity of electrochemical response derived from the current of ferri/ferrocyanide redox indicator that was repulsed from the electrode surface. Further electrode incubation with potassium ions led to the formation of a G-quadruplex that was distinguished with an even higher charge density that caused a more pronounced repulsion of ferri/ferrocyanide redox couple and a smaller current response. The peak current decrease was linear with the logarithm of potassium ions concentration in the range of 10 pM–0.1 µM and 0.5 mM to 1 mM with a LOD of 0.13 pM. No significant signal change was observed after electrode incubation with 10 μM Ca^2+^, Mg^2+^, Na^+^, Li^+^, NH_4_^+^, Rb^+^, Cs^+^, Sr^2+^ and Ba^2+^[48]. 

#### 4.1.2. Optical Potassium Ion Aptasensors

Aside from multiple examples of electrochemical aptasensors dedicated to potassium ion detection, numerous biosensors were also developed with optical signal readout. In such an approach, gold nanoparticles can be used. Chen et al. applied them to immobilize thiolated aptamer strands that protected the nanoparticles from aggregation (Figure 3) [49]. In the presence of potassium ions, a random coil structure of aptamer was changed in the compact G-quadruplex structure, enabling further aggregation of nanoparticles that caused a change of color of the solution. The applied sequence was as follows: 5′-TTTGGTTGGTGTGGTTGGTTT-3′, and the UV–Vis absorption spectra of the solution altered upon the addition of potassium ions. The absorbance at 520 nm decreased, allowing for the detection of K^+^ ions in the range from 5 nM to 0.1 mM with a LOD of 5 nM. The absorbance did not change in the presence of interfering ions such as Ca^2+^, Mg^2+^, Na^+^, Li^+^, Fe^2+^, Zn^2+^, and NH_4_^+^, which proved the high selectivity of the proposed aptasensor. Potassium ions concentration was also detected in real samples (urine) spiked with 1 µM K^+^ ions, and the recovery reached 94–103%. 

The same group elaborated on a similar system. In this approach, AuNPs were applied, and on their surface, unmodified aptamer strands were adsorbed [50]. In this example, the nanoparticles were initially prevented from aggregation by the presence of citrate ions. Additionally, NaCl (40 mM) was added to the solution; however, the aggregation was not happening since the absence of potassium ions made the aptamer strand a random coil structure. After adding potassium ions, a G-quadruplex structure was formed, allowing for nanoparticles aggregation. This caused the change of solution color that could be observed with the naked eye. Such a system enabled potassium ion detection in the range from 1 μM to 1 mM with a LOD of 0.42 nM. Little UV-Vis spectra change was observed in the presence of 1 mM Ca^2+^, Mg^2+^, Na^+^, Hg^2+^, and NH_4_^+^, which proved the aptasensing system selectivity. Finally, the potassium ions were detected in real samples—100-fold diluted urine. The results were in agreement with atomic absorption spectroscopy (AAS) measurements.

Gold nanoparticles were also used by Naderi et al. [51]; however, this time together with a cationic dye—Cationic Yellow. In the proposed system, there was no need for covalent immobilization of aptamer strands on the surface of gold nanoparticles and the addition of salts. The aptamer was immobilized on the gold surface thanks to the presence of nitrogen atoms and exocyclic amino and keto groups in DNA. After the addition of the dye, a color shift was observed with the naked eye. At first, gold nanoparticles modified with aptamer and incubated with the dye showed orange color, and two UV-VIS peaks at 420 and 530 nm were recorded as a result of mixing two colors—red-colored gold nanoparticles and yellow dye. Then, after the introduction of potassium ions, the binding between K^+^ ions and aptamer led to the separation of aptamer from the surface of nanoparticles. The addition of dye caused a solution color change from orange to green, resulting from a mixture of the blue color of aggregated nanoparticles and the yellow dye. This was also evidenced by the absorbance decrease at 530 nm and increase at 700 nm. The proposed sensor enabled potassium ions detection with a LOD of 4.4 nM. It was also shown that the presence of interfering ions such as Ni^2+^, Fe^3+^, Ba^2+^, Ca^2+^, Cu^2+^, Zn^2+^, Hg^2+^, and Mg^2+^ did not lead to the displacement of aptamer strand from nanoparticle surface even at 10 mM concentration. Finally, the elaborated system was transferred to a paper device and allowed for potassium ion determination in the linear range from 10 µM to 40 mM with a LOD of 6.2 µM.

Another approach in the development of optical potassium aptasensor was shown by Verdian-Doghaei and co-workers [52]. This time, potassium ions were detected using a 30-nucleotide strand known as insulin binding aptamer (IBA). Such a sequence could form an intramolecular parallel G-quadruplex structure. In this approach, a triple—helix structure was formed with a central IBA sequence flanked by dual-labeled oligonucleotide conjugated with FAM fluorophore and BHQ1 quencher molecules (STM) (Figure 4). When the IBA sequence was not present, FAM and BHQ1 were in close proximity to each other. After the introduction of IBA, it bound to the loop sequence of STM thanks to the Watson Crick and Hoogsteen base pairing, which increased the FAM signal’s intensity. After adding potassium ions, IBA formed a G-quadruplex structure, and STM oligonucleotide was released. This caused the change of oligonucleotide conformation from linear to loop structure, and as a consequence, a smaller signal was obtained. A fluorescent signal was detected at 520 nm, and a linear response range was achieved from 0.05 to 1.4 mM with a LOD of 0.014 mM. A small response was obtained for interfering cations at 1 mM concentration, including Na^+^, Li^+^, NH_4_^+^, Ca^2+^, and Mg^2+^.

In another fluorescence approach, a switch-on aptasensor was elaborated for potassium ions detection [53]. For that purpose, two aptamer strands were applied, namely 5′-GTGGGTAGGGCGGGTTGGACCACACCAACC-3′ (aptamer 1) and 5′-TGAGGGTGGGGAGGGTGG GGAA-3′ (aptamer 2). An additional fluorescent label—berberine, was applied, and initially, it exhibited low fluorescence signal at 530 nm. In the presence of potassium ion, a G-quadruplex structure was formed that enhanced the binding of berberine with aptamer strand, which resulted in the increase in fluorescent signal. 20 μM berberine was applied in the studies, and the fluorescence was recorded after 3 min. of solution equilibration. For the first aptamer, the concentration of 0.05 μM was chosen. A strong interaction between berberine and aptamer 1-K^+^ system was confirmed by the binding constant of 1.4 × 10^7^ L/mol, and between a G-quadruplex and berberine was 2.7 × 10^7^ L/mol. The proposed system enabled potassium ion detection in the range from 0–1600 μM with a LOD of 31 nM. For the second aptamer, the binding constant between single strand K^+^ ion-aptamer 2 and berberine was 1.1 × 10^7^ L/mol, and between a G-quadruplex and berberine was 1.9 × 10^7^ L/mol. The proposed system allowed for potassium ion detection in the range from 0–400 μM with a LOD of 100 nM. This showed a better performance of 30-nucleotide aptamer. Small system response in the presence of interfering ions and molecules, including Na^+^, Mg^2+^, Ca^2+^, NH_4_^+^, Cl^−^, CO_3_^2−^, and glucose, was observed at the concentration of 250 and 2500 µM. The proposed sensor was also successfully applied for potassium ions detection in diluted blood serum samples.

In addition, the thrombin-binding aptamer’s dual-labeling (FAM and TAMRA) was applied in fluorescence potassium detection [54]. In the presence of potassium ions, a fluorescent resonance energy transfer (FRET) was observed. To limit the nonspecific interaction of calcium and magnesium ions, EDTA disodium salt was added to the solution. In the absence of potassium ions, a fluorescence signal was observed at 516 nm; in contrast, hardly any signal at 578 nm was detected. After the introduction of potassium ions, the FAM signal at 516 nm decreased, while at 578 nm referring to TAMRA increased, which was the result of conformation change of aptamer strand and induction of FRET. It was shown that the fluorescence intensity ratio defined as R = I_578_/I_516_ increased linearly within the range from 0 to 30 mM. Though the system responded in the presence of sodium ions, the slope of the curve was 10 times smaller than in the case of potassium ions. It was also evidenced that the aptasensor responded to the calcium and magnesium ions, for which the slope of the linear response range was 24 times better than for potassium ions. This influence could be minimized by adding a chelating agent—EDTA disodium salt. Finally, the proposed system was utilized for potassium ions detection in simulated serum and serum extracted from a newborn calf. The latter was realized by the addition of EDTA-2Na.

### 4.2. Mercury Ion

Mercury and its divalent cation are known as toxic pollutants that are dangerous to the environment and human and animal health. It was shown that Hg^2+^ and its forms could accumulate in various organs, and hence, contributing to diseases, including brain, kidney, and liver disorders. Its concentration in the environment increases together with the development of industrial activities [55]. The US EPA defined the maximum contamination level of mercury ions in drinking water at 10 nM [56]. Mercury ion exhibits affinity for thymine, a component of DNA nucleobases. As such, Hg^2+^ binds to thymine to form so-called thymine bridges between two DNA strands. This phenomenon often entails conformational changes and thus is very often harnessed when designing DNA sensors for mercury detection [57].

#### 4.2.1. Electrochemical Aptasensors

An aptasensor characterized by a simple receptor layer design for impedimetric detection of Hg^2+^ was developed by Gan et al. [58]. A gold working electrode in the form of elongated nanostructures (so-called nanobands) was employed as the platform for the effective immobilization of thiolated Hg^2+^ aptamers. The selective interaction with mercury(II) was provided by the ssDNA aptamer sequence rich in thymine nucleobases. (Figure 5). The binding of mercury ions triggered a conformational change of aptamer to Hg^2+^-bridged structure. Such reorganization facilitates electron transfer, affecting the change in the recorded impedance. Thanks to the application of the developed surface of the working microelectrode, the aptasensor reached interesting parameters such as a LOD at the level of 40 pM and a linear range of response from 0.1 nM to 1 μM.

In another example of mercury detection, An et al. [59] used a liquid-ion gated field effect transistor (FET) in the developed Hg^2+^ aptasensor (Figure 6). In this case, the graphene was used at the transducer layer to facilitate the conjugation of aptamers via appropriate linkers: diaminonaphthalene (DAN) and glutaraldehyde (GA). DNA sequences capable of interacting with mercury ion (3′-amine-TTC TTT CTT CCC CTT GTT TGT-C10 carboxylic acid-5′) were covalently immobilized. Thanks to the label-free mechanism, it was possible to obtain a real-time response to variable mercury(II) concentrations as low as 10 pM.

An interesting mechanism of mercury(II) sensing has been proposed by Ma et al. [60]. Aptasensor exploited the gold nanoparticles functionalized with a specific ssDNA aptamer (5′-COOH-CTT CTT CCC CCC CCT TCT TC-SH-3′). Nanoparticles act as a molecular gate, which may reversibly entrap redox indicator- toluidine blue (TB) inside the mesoporous silica nanocontainers (MSNs) structure. The TB detected by differential pulse voltammetry (DPV) was the source of the electrochemical signal. With the assistance of aptamer, modified AuNPs acted as gatekeepers of encapsulated TB molecules. In this state, a negligible electrochemical signal was recorded. The presence of Hg^2+^ caused the formation of a hairpin structure of aptamer on the surface of AuNPs, followed by the breakage of the bond between the aptamer and MSNs. Nanoparticles desorbed from the surface of NH_2_-MSNs, resulting in the Hg^2+^-dependent release of TB molecules (Figure 7). According to the authors, this method exhibited excellent analytical performance, e.g., a linear range from 10 pM to 100 μM with a detection limit at the level of 2.9 pM.

Hwan and Hyun [61] recently proposed a functionalized glassy carbon electrode covered with electrochemically reduced graphene oxide for voltammetric Hg^2+^ aptasensing. A thymine-rich aptamer labeled with methylene blue was non-covalently immobilized on the electrode through π–π stacking via an intermediate layer of graphene oxide (GO) directly reduced by cyclic voltammetry. In this case, as in the example above, a hairpin structure was formed due to T-Hg^2+^-T bridging. The conformational change induced aptamer desorption from the electrode surface, resulting in a decrease in the current signal. Dynamic response to changes in Hg^2+^ concentrations was observed in the range between 1 fM and 100 nM with a detection limit of 0.16 fM and very high selectivity to other ions.

Another recent example of nanomaterial-enhanced electrochemical detection of Hg^2+^ is aptasensor using of carbon paste electrode (CPE) modified with electrospun nanofibers polyethersulfone and quantum dots (NFs–QDs) [62]. The authors described the protocol of mercury detection in fruit juice samples. Due to the presence of an amino group at the 5′ end of the aptamer (probe sequence [5′-(NH_2_)-TTT TTT TTT TAC AGC AGA TCA GTC TAT CTT CTC CTG ATG GGT TCC TAT TTA TAG GTG AAG CTG T-3′]), it can be covalently conjugated to carboxylic groups of QDs. After the incubation of the prepared aptasensor in methylene blue redox marker (to form MB complexes with guanine nucleobases), the aptasensor was immersed in samples containing variable concentrations of Hg^2+^ (Figure 8). The aptasensor signal increased linearly in the concentration range from 0.1 to 150 nM and exhibited a low detection limit of 0.02 nM.The interference studies showed that in the presence of 100 nM Fe^3+^, Cd^2+^, Pb^2+^, Ag^+^, Sn^4+^, Ni^2+^, Mn^2+^, and Cu^2+^ no significant disturbance of the sensor response was observed, with the simultaneous presence of 50 nM Hg^2+^.

Electrochemical mercury aptasensor, characterized by a very simple working principle, was also developed with the use of “Au nanoflowers” [63]. Its mechanism was based on the structural rearrangement of ssDNA aptamer labeled with methylene blue (5′-SH-(CH_2_)_6_-TTCTTTCTTCGCGTTGTTTGTT-MB-3′) caused by interaction with Hg^2+^. DNA probe switched from loose to a rigid hairpin form of DNA duplex. This conformation caused a shortening of the distance between the MB and GCE electrode surface decorated with gold nanoflowers, which resulted in an increase in peak current recorded by means of square wave voltammetry (SWV) (Figure 9). The detection limit of such an aptasensor was 0.062 fM, and a linear range was observed in the concentration window from 1 fM to 1 nM.

#### 4.2.2. Optical Aptasensors

Aptasensors with optical detection for mercury(II) biosensing employ both classical, flat transducers as well as nanomaterial surfaces. However, the most commonly used are thymine-rich aptamers, as was in the case of electrochemical sensors. For example, Xing et al. [64] proposed an optical biosensor, which was based on the formation of a three-component complex of fluorescent dye SYBR GREEN I (SGI), a mercury-specific oligonucleotide (MSO) and Hg^2+^ (Figure 10). MSO aptamer (5′-TTC TTT CTT CCCC TTG TTT GTT-3′) forms a hairpin structure stabilized by Hg^2+^ bridges, which improves the affinity towards fluorescent labels and thus turns on a strong emission. As demonstrated, fluorescent nanoprobe offered a relatively narrow linear range (10–100 nM) and a low limit of detection of 0.68 nM.

Real-time optical mercury(II) determination can be carried out using an aptasensor constructed on a waveguide platform, as presented by Chen et al. [65]. The sensor was based on the competition of the interaction of a fluorescently labeled aptamer (Cy5.5-ATCCCCTTTGTTTGTTTAGCCCCTATTCTTTCTTGGTCTGC) with mercury and a complementary sequence on the surface (GCAGACCAAGAAA). It owed its simple principle to the fact that the probe exhibited a higher affinity for mercury than cDNA. The lack of hybridization resulted in a decrease in the fluorescence intensity excited via the waveguide. (Figure 11). The authors determined the detection limit in the water sample at 0.2 μg/L, and the response linearity was observed from 1.4 μg/L to 240.7 μg/L Hg^2+^.

To further improve the analytical parameters of Hg^2+^ detection by fluorometric aptasensors based on the use of probes containing thymine-thymine mismatches, additional signal amplification strategies must be employed. An example is a sensor described by Zhang et al., where enzyme-assisted amplification by the introduction of nicking endonuclease was used. Thanks to this, the dynamic range was improved from 20 pM to 10 nM, and the detection limit reached 6 pM of Hg(II) [66]. Another application of enzymatic signal amplification and gold nanoparticles in mercury aptasensor construction was proposed by Memon et al. [67]. Authors developed a novel concept of optical aptasensor for sensitive, colorimetric Hg^2+^ detection using bare gold nanoparticles (AuNPs) and Exonuclease-III (Exo-III) as signal amplifiers and Co^2+^ as a cofactor.

Recently, an electrochemiluminescence (ECL) -based biosensor with signal amplification has also been described for the optical detection of mercury ions at ultra-low levels. In the proposed approach, the recognition of mercury by thymine-rich sequences triggered rolling circular amplification (RCA) (Figure 12). RCA products were detectable by ECL by hybridization with tris (bipyridine) ruthenium (TBR) -tagged probes. The biosensor signal increased linearly in the concentration window 0.1–1000 nM, showed a LOD about 100 pM and excellent selectivity over Mg^2+^, Ca^2+^, Fe^2+^, Fe^3+^, Cd^2+^, Co^2+^, Cu^2+^, Ag^+^, Ni^2+^, Pb^2+^, and Zn^2+^ ionin concentration 1 and 100 μM [68].

Label-free optical aptasensor for Hg^2+^ detection was also developed using Atomic Force Microscopy (AFM), as described by Li et al. [69]. The aptamer sequence (5′-NH_2_–(CH_2_)_6_–TCATGTTTGTTTGTTGGCCCCCCTTCTTTCTTA-3′) was conjugated to the AFM probe, and the adhesion force between the probe and a flat graphite surface was measured by single-molecule force spectroscopy (SMFS). The addition of mercury ions induced a change in adhesion force. The detection limit reached 100 pM, and the linear range was from 10 to 1000 pM. Such aptasensor exhibited very high selectivity for Hg^2+^ over other metal cations at the same concentration (1μM), such as K^+^, Ca^2+^, Zn^2+^, Fe^2+^, and Cd^2+^.

### 4.3. Silver Ion

Silver(I) cation is a highly bioactive compound, which exhibits a strong antibacterial effect at low concentrations, but at a high level, it exerts biological effects on human health. It can potentially accumulate in the human body via the food chain, causing argyria and damaging the brain, immune, and nerve systems [70]. The maximum contamination of silver ions in water, according to EPA, is 0.1 mg/L [71]. Single-stranded DNA fragments and mismatch regions of dsDNA rich in cytosine are the most widely used in sensor applications for Ag^+^ ion detection.

#### 4.3.1. Electrochemical Aptasensors

Zhang et al. [72] proposed a simple electrochemical Ag^+^ aptasensor working in a sandwich assay format. Both the surfaces of the gold electrode and gold nanoparticle label were functionalized with a cytosine-rich ssDNA. In the presence of silver ions, C-Ag^+^-C structures bridged DNA immobilized on the electrode (5′–CCTCCAACCTCT–(CH2)_6_–SH–3′) and DNA modified AuNPs, which caused sandwich-like structure formation. In order to increase the current signal, the authors used silver enhancement, which involved the deposition of silver on the gold label surface (Figure 13). In this process, silver ions (from an external source) were reduced to metallic silver by using a reducing agent, and the electrochemical signal derived from the electrooxidized silver was detected voltammetrically. The limit of detection was 2 pM, and the linear range was from 5 to 50 μM. This sensor was characterized by high selectivity, especially for Na^+^, K^+^, Ba^2+^, Mg^2+^, Zn^2+^, Pb^2+^, Mn^2+^, Co^2+^, Ni^2+^, Fe^2+^, Fe^3+^, Al^3+^ (all at 5 μM), which induced negligible signal change, as well as their presence, does not negatively affect the signal coming from Ag^+^.

Another example of electrochemical detection of silver ions using an impedimetric aptasensor was proposed by Gong and Li [73]. This simple sensor comprised Y- shaped, C-rich double-stranded DNA in the receptor layer (Figure 14). Ag^+^ is able to bridge C-C mismatches to form C-Ag^+^-C complexes and thus change conformation on the DNA probe. This was reflected in the change of charge transfer resistance. After the addition of Ag^+^, the free C-rich part in dsDNA underwent a spatial rearrangement to form an intramolecular C–Ag^+^–C duplex and increased the film thickness. The resistance values for the films after and before the addition of other metal ions (such as Zn^2+^, Co^2+^, Ni^2+^, Mn^2+^, Cd^2+^, Cu^2+^, Al^3+^, Mg^2+^, Li^+^, Ca^2+^, Fe^3+^, Hg^2+^, and Pb^2+^) with the same concentration (10^−5^ M) instead of Ag^+^ demonstrated that all selected interferents gave a negative resistance shift and Ag^+^ gave a positive response. The low detection limit of this sensor towards Ag^+^ was 10 fM.

The next example of electrochemical detection of silver ions is a ssDNA oligonucleotide-based sensor proposed by Kim [74]. The presence of Ag^+^ influences the conformation of the DNA strand and thus changes the electron transfer efficiency. At the beginning, the aptamer was in the form of single-stranded DNA with the methylene blue (MB) labeled to one of the ends of the DNA sequence, located far from the electrode. In the presence of silver ions, the DNA changes its structure to z hairpin probe DNA. This is possible due to the high affinity of silver ions to cytosine and the formation of C-Ag-C structures. It caused methylene blue to be near the electrode surface, which allowed for easier electron exchange and, thus, the increase in voltammetric signal (Figure 15). This sensor had a LOD at the level of 10 nM with a linear range between 10 and 200 nM. The behavior of the aptasensor after the addition of other metal ions such as Mg^2+^, Ca^2+^, Ba^2+^, Fe^3+^, Co^2+^, Ni^2+^, Cu^2+^, Zn^2+^, Cd^2+^, and Pb^2+^ at 500 nM represented by lack of signal change confirmed good selectivity of the developed approach.

#### 4.3.2. Optical Aptasensors

Optical detection of silver ions can be carried out using a fluorometric sensor based on Exonuclease I activity modulation as proposed by Wei et al. (Figure 16) [75]. The working principle involves a specific interaction of Ag^+^ and cytosine-cytosine base mismatch. In this approach, the authors used berberine (Ber) as a fluorescent marker. Ber/Ag^+^-aptamer complex can be easily digested enzymatically, resulting in reduced fluorescence. In the presence of silver ions, the aptamer (5’-CCT CCT CCC TCC TTT TCC ACC CAC CAC C-3’) formed C-Ag^+^-C bridges and changed its structure so that it became resistant to degradation. For this sensing mode, the detection limit of Ag^+^ was 4.4 nM, and the linear range was from 0.0059 μM to 235.48 μM.

The next approach to silver(I) optical turn-off aptasensing is the use of SERS substrate with Au@Ag core-shell nanoparticles (Au@AgNP), as proposed by Wu et al. (Figure 17) [76]. The thiolated aptamer terminated with the Raman label (5′-Rox-CTCTCGGATCTTCTCCGTCTTTTTCAACACAAGATCCCACACTTTTTTTTTT-SH-3′) was immobilized on the NP via hybridization with the complementary DNA (5′-GTGTGGGATC-3′) to form the dsDNA. When silver ions were present, the ROX-aptamer was turned into a cytosine C-Ag^+^-C mediated hairpin structure. Rearrangement of the label to the substrate neighborhood due to a conformational change increased the intensity of the Raman signal as the concentration Ag^+^ increased. The detection limit of such an aptasensor was 50 pM, with the linear detection range of 0.1 to 100 nM.

In silver ions detection, a typical analyte-induced aggregation of nanomaterials was also used [77]. The process was induced both by the formation of C-Ag^+^-C bridges [78] as well as by the competitive desorption of the aptamer, which also plays the role of nanoparticles stabilizer [79]. This approach has been exemplified by using gold nanoparticles, as proposed by Xi et al. [77]. The authors developed aptamer-functionalized gold nanoparticles (AuNPs). The two DNA aptamer strands were tethered to the surface of AuNPs. C-Ag^+^-C bridging triggered aggregation of AuNPs in response to silver ions addition, which led to a color shift from red to purple. Such a simple colorimetric probe exhibited a LOD at the level of 0.236 nM and a linear range from 1 nM to 1 μM. It offered high selectivity towards other metal ions such as Cd^2+^, Co^2+^, Cu^2+^, Fe^3+^, Hg^2+^, Mn^2+^, Ni^2+^, Pb^2+^, Zn^2+^, and NO_3_^−^ anions (all at a concentration of 10 μM) compared with 1 μM Ag^+^.

Another application of gold nanoparticles in silver ions sensing was a label-free optical aptasensor proposed by Li et al. (Figure 18) [80]. It relied on the measurement of light intensity by dark-field microscopy. The mechanism based on the aggregation of gold nanoparticles regulated by exo-nuclease III activity was very similar to the mercury(II) ion sensors described above [64,66]. The presence of silver ions bound by an aptamer with the sequence 5′- CCC CCC CGT GGG TAG GGC GGG TTG GAC CCT ACC CAC CCC CCC G-3′ had a stabilizing effect on gold nanoparticles so that they do not aggregate and retain red color, which could be observed under dark-field microscopy. The detection limit reached 39 fM with a linear range from 57 fM to 57 nM. The selectivity was examined by adding to the solution 20 nM of Co^2+^, Fe^3+^, Mn^2+^, Ni^2+^, Pb^2+^, and Ag^+^, and color change was not observed. Silver ions optical aptasensing was also conducted by tracking the fluorescence anisotropy reduction in G-rich oligonucleotide, as proposed by Zhang and Wang [81]. The change in fluorescence anisotropy allowed the selective detection of Ag^+^ with a LOD of 0.5 nM and a dynamic range from 2.0 to 100 nM.

Recently, an interesting method of silver ions sensing was proposed by Pavadai et al. [82]. The authors constructed a “turn on” biosensor based on Ag-dependent fluorescence restoration. A metal-organic framework containing cobalt (Co-MOF) was used as a quencher. Three cysteine-rich aptamers labeled with the FAM fluorescent label were adsorbed on MOF to decrease the background signal. The introduction of Ag^+^ ions into the environment resulted in the formation of bridges between the strands, which triggered their spatial rearrangement from the form of separate hairpins to connected, double-stranded structures characterized by a triangle shape. Desorption of the DNA from the surface resulted in an Ag^+^-dependent increase in the fluorescence signal. The limit of Ag^+^ detection was 45 pM, with a linear range of 0 to 0.8 nM. This sensor had proven good selectivity towards Ag^+^ over other challenging metal ions (such as Mn^2+^, Mg^2+^, Co^2+^, Ni^2+^, and Hg^2+^).

### 4.4. Copper Ion

Copper ions are an essential micronutrient in our organisms, which plays an important role in many biological processes [83]. On the other hand, copper ions can be toxic, especially at high concentrations. It can cause many diseases such ashemolysis, Wilson’s disease, and in some cases, even death[84]. According to the EPA, the maximum copper contamination in water is 1.3 ppm [71].

#### 4.4.1. Electrochemical Aptasensors

One of the approaches to copper(II) ions detection is the method proposed by Chen et al. [85]. The authors presented an electrochemical aptasensor for Cu^2+^ detection based on gold nanoparticles (AuNPs) assembly on the working electrode. Two DNA sequences were used. First: 5′-GAA TTC TAA TAC GAC TCA CTA TAG GAA GAG ATG GCG ACT GTT TAG AAG CAG GCT CTT TCT TAT GCG TCT GGG CCT CTT TTT AAG AAC-3′ was attached to the AuNPs surface. The second, Ferrocene (Fc)-labeled: 5′-Fc-(CH_2_)_6_-GAATATAGTGAGCTA CGC TAG AAT TC-(CH_2_)_6_-SH-3′ was the source of the electrochemical signal detected by the square wave voltammetry (SWV). Both sequences were partially complementary to each other. In the absence of Cu^2+^ (Figure 19, on the left), the double helix was rigid enough to prevent the redox label from getting close to the electrode, and the recorded current signal was small. After copper binding by the aptamer, the structure was partially unraveled, and the released strand could allow ferrocene to approach the electrode. Therefore, an increase in signal was observed in the presence of copper(II) over the range from 0.1 nM to 10 μM, with a detection limit of 0.1 pM. Good selectivity was confirmed against other divalent metal ions (1 μM Ca^2+^, Co^2+,^ Mg^2+^, Ni^2+^, Pb^2+^, Sn^2+^, and Zn^2+^). These metals were not found to interfere with the detection of Cu^2+^.

The next example of copper ions aptasensing has been proposed by Qi et al. (Figure 20) [86]. It was based on a disposable gold-plated coplanar electrode array covered with aptamer for Cu^2+^ (ATC GCG ATA TTT TCT GTA GCG ATT CTT GTT TGA GCG CTC GGT ACG AAC AGA). The detection was carried out by capacitance measurements. To further improve the sensitivity by concentrating the analyte, an alternating-current electrothermal effect was additionally engaged. This simple sensor on the base of the monolayer of receptors on a gold electrode allowed achieve a limit of detection of 2.97 fM and responded linearity from 5.0 fM to 50 pM. Some common cations were examined in terms of sensor selectivity, e.g., Fe^2+^, Pb^2+^, As^3+^, As^5+^, Hg^2+^, Mg^2+^, Al^3+^, Cr^3+^, and Cr^6+^. As shown, interferents at concentrations of three orders of magnitude higher than Cu^2+^ did not induce a significant change in the aptasensor signal.

Another example of Cu^2+^ detection is a label-free biosensor based on a graphene field-effect transistor described by Wang et al. [87]. The working principle takes advantage of the change in surface charge (and thus the potential of the gate modified with aptamer sequence 5′ -HS-S-(CH_2_)_6_- ATC GCG ATA TTT TCT GTA GCG ATT CTT GTT TGA GCG CTC GGT ACG AAC AGA -3′) due to a change in its conformation upon copper ions binding. The observed concentration-dependent response to the target ions was a decrease in recorded gate potential. As constructed, the transistor-based aptasensor offered a low detection limit of 10 nM and a linear range of 10 nM to 3 µM while maintaining high specificity—a number of ions at concentrations 30-fold higher than Cu^2+^ showed only negligible effects on the recorded potential.

#### 4.4.2. Optical Aptasensors

The use of AS1411 aptamer (26-mer G-rich ssDNA sequence) is an example of the application of an already existing aptamer sequence in copper ions detection. This aptamer is known for its affinity to nucleolin protein occurring on the surface of cancer cells. Bahreyni et al. [88] discovered that this aptamer could also bind copper ions. Biotinylated magnetic nanoparticles modified with aptamer AS1411 (5′-Biotin-GGTGGTGGTGGTGGTGGTGGTGGTGGTGGTGGTGGTGGTGGTGGTGGTGG-3′) served to capture and concentrate the analyte. The use of GelRed as a fluorescent dye showing an affinity for dsDNA allowed the construction of a “turn-off” aptasensor (Figure 21). The detection limit of the sensor was 0.01 μM towards Cu^2+^ in serum samples. The selectivity of fabricated Cu^2+^ aptasensor towards common cations (2 μM Mg^2+^, Zn^2+^, K^+^, Fe^3+^, Cu^2+^, Ca^2+^, Mn^2+^, Na^+^ and Cr^3+^) was also confirmed.

Not only the DNA sequences themselves but also their complexes with the primary analyte may become receptors capable of selective interaction with copper ions. Wei et al. proposed a colorimetric method for the determination of Cu^2+^ based on the binding of ions through the previously formed K^+^—aptamer complex [89]. The addition of K^+^ influenced the transition of the G-rich aptamer sequence into a G-quadruplex. As obtained structure interacted with copper ions and the obtained K^+^ -aptamer-Cu^2+^ complex had increased peroxidase-like activity, the detection of which was the aptasensor’s working principle. The LOD of Cu^2+^ in an investigated solution was 0.076 μM.

Interaction of DNA probes with Cu^2+^ harnessed in construction in biosensors does not have to trigger conformational changes but also can induce analyte-dependent self-cleavage of oligonucleotides in the receptor layer. Such receptors, called DNAzymes, can easily generate analyte-dependent signals, both electrochemical [90,91] and optical [92], as a result of cleavage sequence at a well-defined location. For example, Huang et al. [93] designed a molecular beacon terminally labeled with a fluorophore, which in its initial form did not emit fluorescence due to the quenching by the neighboring graphene oxide. The copper ion-induced cleavage of the sequence released the fluorophore, which was the working principle of the fluorescent “turn on” nanoprobe. The detection limit of Cu^2+^ was about 50 nM.

### 4.5. Lead Ion

Lead ions are toxic metal pollutants that are dangerous for human and animal health. It was shown that lead ions could accumulate in various organs and hence can cause diseases, including brain, kidney, and liver disorders. The US EPA defined the maximum contamination level of lead ions in drinking water as 72 nM. That is why it is crucial to elaborate on a tool for Pb^2+^ cations that would be sensitive, simple to use, and cheap. One of the solutions is the application of aptamer strands as the receptor for lead ions detection. In such cases, guanine-rich sequences are applied as lead ions can stabilize a G-quadruplex from free- coil sequences. One of the major challenges in the case of electrochemical sensors is to minimize the nonspecific interaction of redox label/indicator with transducer surface as well as limited selectivity, especially in the presence of potassium ions that can also stabilize the G-quadruplex structure [43].

#### 4.5.1. Electrochemical Aptasensors

In one of the electrochemical aptasensor examples, the lead (II) specific aptamer (LSA): 5′-GGG TGG GTG GGT GGG T-C_6_ -SH-3′ was used (Figure 22) [94]. The sequence was covalently immobilized on the gold surface, and this was proceeded by the introduction of the graphene molecules through hydrophobic and π-π interactions into the receptor layer. This was followed by the addition of thionine molecules that interacted with graphene thanks to π-π interactions. After the addition of lead ions, the random-coil structure of the aptamer was changed into a dense G-quadruplex form that caused the release and repulsion of graphene and electrochemically active thionine. This resulted in the reduction in the electrochemical response of the sensor. The high sensitivity of the sensor was provided by graphene molecules, allowing high surface area and high conductivity. The binding ratio between the thionine and aptamer strand was calculated to be 8.1, which improved the sensitivity of the sensor. The 100 min. incubation time of lead ions with the aptamer-modified electrode was chosen on the basis of the optimization studies. The oxidation peaks of thionine decreased with the increase in lead ions concentration from 1.6 × 10^−13^ to 1.6 × 10^−10^ M with a LOD of 3.2 × 10^−14^ M. The aptasensor also showed high selectivity towards lead ions, and the influence of potassium ions was minimized thanks to the higher association force of lead ions towards the aptamer as well as the presence of graphene, which reduced the possibility of aptamer conformation change.

An interesting approach in electrochemical aptasensor development is the application of peroxidase-like GR sequence aptamer—functionalized 3D—flower MoS_2_ microsphere hybrid as signaling probes (Figure 23) [95]. The aptasensor was prepared as follows. Firstly, the surface was modified with a solution of gold nanoparticles and multi-walled carbon nanotubes. The next step was the immobilization of the aptamer strand, which was proceeded by the incubation with the blocking agent—6-mercapto-1-hexanol. Then, lead ion samples were mixed with MoS_2_–GR and incubated with the modified electrode. Finally, a solution containing hydrogen peroxide and TMB was placed on the electrode surface. Under the optimized conditions in the presence of lead ions, the GR sequence was cleaved at the ribonucleotide site, leading to the formation of shorter MoS_2_—oligonucleotide fragments that hybridized with a hairpin probe immobilized on the electrode surface. This allowed sensitive detection of lead ions using the DPV technique in the range from 0.03 to 500 nM with a LOD of 0.015 nM. Hardly any response was detected in the presence of 5 µM cations, including Fe^3+^, Cd^2+^, Co^2+^, Zn^2+^, Mn^2+^, Ni^2+^, Cu^2+^, Hg^2+^, and Ag^+^. The proposed assay was also used for lead ions determination in real samples, such as water samples, fruit juice samples, and solid samples.

Another approach to electrochemical lead ions analysis was presented by Taghdisi et al. [96]. They used enzymes, which digest aptamers (as typical nucleic acid fragments) in the detection mechanism. The assay was composed of a hairpin structure of a complementary strand of aptamer, gold nanoparticles, and exonuclease III. In the absence of Pb^2+^ ions, gold nanoparticles were attached to the gold electrode surface. In the absence of lead ions, hybridization between the aptamer strand and complementary strand that was immobilized on the surface of gold occurred. The addition of Exo III caused the degradation of the 3′ end of the complementary strand and the release of the aptamer strand. The remains of complementary strands bound with gold nanoparticles interact via electrostatic and van der Waals interactions. This allowed for a strong electrochemical signal in the presence of a ferri/ferrocyanide redox indicator. In contrast, the addition of lead ions caused the binding between the aptamer and lead ions and released the aptamer from the complementary strand. The complementary strand formed a hairpin structure that cannot be cleaved by an exonuclease and also does not bind with gold nanoparticles. Moreover, the negative charge of the complementary strand caused the repulsion of a ferri/ferrocyanide redox indicator and a decrease in the current signal. Such an approach allowed lead ions detection in the range from 0.7 to 300 nM with a LOD of 149 pM. The proposed assay also showed high selectivity towards lead ions and was negligible for analyzed interfering compounds.

A field-effect transistor-based aptasensor toward lead ions was also elaborated [97]. FET was modified with single-wall carbon nanotubes that were further modified with a duplex structure formed of aptamer with the abundance of guanine moieties and complementary strand that were immobilized due to the presence of amine groups. Upon the addition of lead ions, a G-quadruplex structure was created that led to the release of the aptamer strand from the complementary sequence. This caused a change in the conductivity of SWNTs. The relative resistance at 0.02 V increased linearly with lead ions concentration in the range from 1 ng/L to 100 µg/L with a LOD of 0.39 ng/L. The relative resistance did not vary upon FET incubation with interfering ions, which indicated the high selectivity of the proposed sensor toward Pb^2+^ ions.

#### 4.5.2. Optical Aptasensors

There are several examples of fluorescence lead ion aptasensors. Their detection mechanisms are similar to those presented in previous examples for potassium ions. However, a completely different approach was presented by Verma et al. [98]. In this case, the lead ions were detected using a “spinach aptamer probe” and different orientations of liquid crystals (LCs). In such case, N,N-dimethyl-n-octadecyl-3-aminopropyltrimethoxysilyl chloride (DMOAP)-modified glass slides were incubated with liquid crystals that led to the homeotropic orientation of the liquid crystal layer in the presence of cationic surfactant cetyltrimethylammonium bromide (CTAB). This resulted in the dark polarized optical image of liquid crystals. The orientation of LC switched to planar upon the interaction of CTAB with spinach RNA sequences, leading to the bright orientation of the liquid crystals. Finally, the addition of lead ions enabled their binding to the aptamer strand, reorientation of CTAB, and further change of position of LC to homeotropic that caused a transition of the liquid crystals’ appearance to dark. The proposed aptasensor allowed achieve a lower limit of detection of 3 nM. It was also observed that the LC sensor was bright in the presence of 75 nM Co^2+^, Hg^2+^, Ni^2+^, and Zn^2+^ during 20 min of observation, whereas it changed to dark within 5–10 min. after the addition of 75 nM of Pb^2+^.

### 4.6. Cadmium Ion

Cadmium ions are one of the most harmful metal ions that can accumulate in the human organism. This might cause different diseases, including renal dysfunction, several types of cancer, as well as bone degeneration. US EPA defined the maximum level of cadmium in drinking water as 5 μg/L. One of the possibilities for cadmium ions detection is the utilization of aptamer-based layers [99].

#### 4.6.1. Electrochemical Aptasensors

There are several examples of cadmium ions detection using glassy—carbon electrodes modified with carbon-based nanomaterials. Chang-Seuk et al. [100] used such an approach with GCE modified with electrochemically reduced graphene oxide, and additionally with a duplex structure formed of the Cd aptamer modified with methylene blue and the complementary DNA strand: 5′-GGGGGGGGACTGTTGTGGTATTATTTTTGGTTGTGCAGT-methylene blue-3′ (aptamer) and 3′-TGACAACACCATAATAAAAACCAACACGTCA-5′ (cDNA). The binding of the duplex was possible due to the π-π interaction between guanine ending at the 5′ end and the surface of the electrochemically reduced graphene oxide. The formation of the receptor layer on the electrode surface was confirmed using the impedance spectroscopy technique, and differential pulse voltammetry was applied as a detection technique. Firstly, the formed duplex structure did not allow for efficient charge transfer between the methylene blue label and the electrode surface. Upon the introduction of cadmium ions, the complementary strand was separated from the aptamer strand, and the aptamer underwent the conformation change, limiting the distance between the redox label and the electrode surface; hence, the current response increased. The current signal increased within the range of 1 fM to 10 nM, with a LOD of 0.65 fM. The current response increased substantially only in the presence of cadmium ions, whereas the addition of Zn^2+^, Pb^2+^, Mn^2+^, Co^2+^, Fe^2+^, and Cu^2+^ did not lead to pronounced signal change. The increase in current was recorded after mercury ions addition due to the presence of thymine nucleobases in the receptor layer.

In another approach, a glassy carbon electrode was modified with carbon nanotubes in chitosan solution (Figure 24) [101]. Subsequently, gold nanoparticles were electrodeposited, and the electrode was immersed in chitosan solution. That was followed by immersion in glutaraldehyde solution, which was proceeded by incubation with amine-terminated aptamer. The performance of the aptasensor was verified with the use of impedance spectroscopy in the presence of a ferri/ferrocyanide redox indicator. The charge transfer resistance increased in the range of concentration between 10^−13^ to 10^−4^ M. Thecharge transfer resistance did not change significantly after electrode incubation with interfering ions, namely Hg^2+^, Pb^2+^, and Zn^2+^.

#### 4.6.2. Optical Aptasensors

Fluorescent detection of cadmium ions was performed, e.g., by Luan et al., with the application of a PicoGreen double-stranded DNA label (Figure 25) [102]. The assay functioned as follows: firstly, the aptamer was incubated with cadmium ions, which caused their conformation change. Then, the complementary strand was added to the solution along with the Pico Green label that interacted with the double-stranded DNA that was formed by free aptamer strands and the complementary sequence. The fluorescence signal decreased with the increase in Cd^2+^ concentration in the range from 10^−10^ to 10^−4^ g/mL. The fluorescence intensity did not vary after the addition of various concentrations of interfering metal cations. The proposed sensor was also utilized for the detection of cadmium ions in real samples, such as lake water.

### 4.7. Arsenic Ion

Arsenic ions are considered one of the most toxic pollutants that occur in the environment. They are known to pose a threat, especially in terms of cancer diseases. Arsenic ions exist in two forms: trivalent and pentavalent ions, though the trivalent ions are known to be more toxic and more water-soluble. US EPA defined the safe level of arsenic ions to be 10 μg L^−1^ or 133 nmol L^−1^. One of the approaches to arsenic ions detection is the application of aptamer-based sensing layers [103].

#### 4.7.1. Electrochemical Aptasensors

For electrochemical detection of arsenic ions, except aptamers, also nanomaterials were used. A 42 GT thiolated ssDNA sequence was immobilized on the gold disk electrodes that was followed by incubation with 6-mercapto-1-hexanol used as a blocking agent. Then, graphene oxide was deposited, further enabling the in situ generation of Prussian blue nanoparticles that were applied as a label and a source of the electrochemical signal (Figure 26) [104]. In the presence of arsenite ions, the complex formation between ssDNA and arsenite occurred, which led to the adsorption of less graphene oxide followed by fewer Prussian blue nanoparticle generation. As a result, a smaller current signal was recorded. The effectiveness of each modification step was confirmed using impedimetric techniques in the presence of a ferri/ferrocyanide redox indicator. The optimized conditions for biosensor preparation were as follows: 4 µM ssDNA, 30 µM PB precursor solution, 60 min. graphene oxide adsorption, 30 min. for PB nanoparticles generation as well as pH 7.5, and 3 h incubation period with arsenite. In such an approach, it was possible to detect arsenite in the range from 0.2 to 500 ppb, with a lower limit of detection of 0.058 ppb. A pronounced interference was caused by mercury ions due to the presence of thymine nucleotides. Such an occurrence was minimized by the addition of the chelating agent EDTA. The elaborated sensor was also applied to unspiked and spiked tap water, river, and lake samples.

In order to make arsenic ions detection more convenient from a practical point of view, screen-printed electrodes were used as miniaturized transducers. The first step of biosensor preparation was the electrodeposition of gold nanoparticles on the surface of the electrode, followed by the immobilization of thiolated arsenic aptamer. This was followed by the incubation with arsenic ions and graphene oxide- methylene blue composite. The quantitative studies were performed using differential pulse voltammetry (DPV). The peak currents decreased with increased arsenic ions concentration in the range from 4 × 10^−4^–10 mg L^−1^, with a LOD of 2 × 10^−4^ mg L^−1^. The proposed sensor showed good selectivity towards arsenic ions and allowed analysis of standard reference material as well as in real shellfish samples [105].

A label-free “signal on” electrochemical sensor was proposed by Cui et al. (Figure 27) [106]. In this approach also screen—printed electrodes were used. The carbon electrodes were modified with gold nanoparticles followed by immobilization of thiolated arsenic aptamer. Cationic polydiallyldimethylammonium (PDDA) was used to minimize the negative aptamer’s charge, which also led to the adsorption of fewer ruthenium hexamine redox indicator molecules. After the addition of arsenic ions, they formed complexes with aptamer strands that led to the conformation change. This caused smaller adsorption of PDDA and attraction of ruthenium hexamine, which resulted in a current signal increase. The proposed assay allowed arsenite detection in the range from 0.2 nM to 100 nM with a LOD of 0.15 nM. The aptasensor showed high selectivity towards arsenite ions even when a 5-fold higher concentration of interfering ions was introduced.

#### 4.7.2. Optical Aptasensors

An interesting approach dedicated to arsenic ions determination was presented by Nguyen et al. [107]. Trivalent ions were detected with the use of liquid crystals in the presence of a specific aptamer and a cationic surfactant cetyltrimethylammonium bromide (CTAB), which formed a self-assembled monolayer that allowed for homeotropic anchoring of liquid crystals. In the absence of arsenic ions, the aptamer strands caused the disruption of self-assembly of CTAB due to electrostatic interaction between the CTAB and aptamer, causing the planar orientation of liquid crystals. Then, the complex between the aptamer and arsenic ions was formed in the presence of arsenic ions, leading to the aptamer conformation change. Consequently, the interaction between the aptamer and CTAB was weakened and caused the homeotropic orientation of liquid crystals. Such an approach allowed obtaining a lower limit of detection of 50 nM. The optical appearance of the liquid crystals changed within 30 min. from dark to bright after incubation of 150 pM aptamer with 500 nM Ca^2+^, Co^2+^, Ni^2+^, Fe^2+^, Zn^2+^, Pb^2+^, and Hg^2+^, while it remained dark after incubation with 500 nM arsenic ions. The proposed assay was also used for the analysis of tap water spiked with arsenic ions.

Moreover, the not obvious approach for fluorescent detection of arsenic ions was presented in [108]. The triple-helix molecular switch (THMS), exonuclease III, and 2-amino-5,6,7-trimethyl-1,8-naphthyridine (ATMND) were used in the assay. THMS contained a label-free signal transduction probe and a hairpin-shaped aptamer strand with two arm segments. A fluorescent label ATMND bound to cytosine at C-C mismatch thanks to π- π interactions. A hairpin probe consisted of three sections, with a C-C mismatch at the stem so that ATMND/hairpin probe complex could be formed. The other two segments at the 3′ and 5′ allowed to recognize the signal transduction probe, and a secondary target analog enclosed in the stem for further amplification. Exo III was applied to recycling both initial and secondary signal transduction probes. In the absence of arsenic ions, THMS was switched off, and the signal transduction probe was locked. As a result, the enzyme could not digest ATMND/hairpin complex with more than 4 bases overhang at the 3′ end. As a result, the complex was very stable and exhibited low fluorescence. Then, after the addition of arsenic ions, THMS was switched on, releasing a signal transduction probe which allows for hybridization of ATMND/hairpin with signal transduction probe that allows for Exo III digestion, followed by the release of signal transduction probe, a secondary STP analog, and ATMND. The cycle was repeated with the STP initial and secondary hybridizing with other complexes that yield free ATMND and thus lead to an increase in fluorescence. The emission spectra recorded at 408 nm increased in the range from 10 ng/L to 10 mg/L with a LOD of 5 ng/L. The high selectivity towards arsenic ions was confirmed as the fluorescence intensity did not change significantly upon the addition of As^5+^, Ag^+^, Cd^2+^, Cu^2+^, Mg^2+^, Mn^2+^, Ni^2+^, Pb^2+^, Zn^2+^, tetracycline (TE), and methyl parathion (MP).

Another fluorescence assay was elaborated by Ensafi et al. [109]. In this approach, a specific aptamer was immobilized on the surface of cysteamine—stabilized CdTe/ZnS core/shell quantum dots, which led to the aggregation of quantum dots and quenching of the fluorescence. After the addition of arsenic ions, the aptamer strands formed complexes with arsenic ions that caused degradation of quantum dots and increased the fluorescence. The proposed assay showed a dynamic response in the range from 1.0 × 10^−11^ to 1.0 × 10^−6^ mol L^−1^, with a lower limit of detection of 1.3 pmol L^−1^.

The easy and convenient colorimetric detection of arsenic ions was also performed using silver nanoparticles (Figure 28) [110]. Silver nanoparticles were modified with thiolated aptamer, which further interacted with arsenic ions. This led to the decrease in absorbance peak at 403 nm. Such an approach allowed for arsenic ions detection in the range from 50 to 700 µg L^−1^, with a lower limit of detection of 6 µg L^−1^.

For arsenic ions detection, a dual optical and electrochemical sensor was also elaborated (Figure 29) [111]. For that purpose, a glassy carbon electrode was modified with chitosan–Nafion layer, followed by the introduction of a glutaraldehyde crosslinker that provided binding with amine-modified DNA. Then, BSA was used as a blocking agent. The arsenic-specific aptamer was hybridized with a complementary DNA immobilized on the surface. Finally, the CNT-BSA nanocomposite was attached to the aptamer through glutaraldehyde. Arsenic ions detection was performed in the presence of a ferri/ferrocyanide redox indicator. The proposed aptasensor showed two linear ranges of response: from 1 to 50 nM and 100 to 500 nM, with a LOD for the former range of 0.78 nM. The aptasensor response was enhanced using CNT-BSA nanocomposite that provided two linear ranges of response from 0.15–10 and 20–100 nM, with a LOD of 0.074 nM. The optical response was also recorded as a strong peak at 255 nm, referring to CNT_COOH_ absorption that decreased upon the addition of arsenic ions due to detachment of CNT-BSA from the receptor layer.

## 5. Simultaneous Detection of Different Metal Cations

Except for examples of dual (optical and electrochemical) sensor elaboration, it would be more beneficial if a defined biosensor could allow for the detection of more than one analyte. This is because most of the aptasensor examples are aimed to be used for complex samples such as river or lake water. Hence, some of the elaborated biosensors for multiple metal ion detection will be presented in the next part of the article. Similarly, as in the examples of biosensors dedicated to single ion detection, also in simultaneous detection of more than one ion, the assays can be divided into those with the optical or electrochemical analytical signal.

One of the optical approaches is the application of an aptamer-based layer for dual detection of mercury and silver cations [112]. For that purpose, a paper was used as a substrate for the elaboration of the receptor layer with two detection zones for silver and mercury ions. For mercury ions, the thymine-based ssDNA was used: 5′-FAM-TTT TTT TTT TTT-3′, and for silver ions detection, 5′-FAM-CCC CCC CCC CCC-3′ was applied. After the injection of DNA strands into detection zones, graphene oxide was added to quench the fluorescence of the labeled ssDNA sequences as it was bound through Π-Π stacking interactions. When the sample was placed in the middle zone, it diffused towards detection zones, where the binding of silver or mercury ions led to the formation of a complex and conformation change that allowed to separate DNA sequence from graphene oxide. In the case of mercury ions detection, the dithymine bridges were formed, whereas, for silver ions, dicytosine bridges were created. This resulted in the recovery of the fluorescence signal. Due to the content of the receptor layer, there was no cross-reactivity of the ions with receptor layers. The proposed sensor allowed mercury ion detection in the range from 0.05 to 50 nM with a LOD of 1.33 pM and silver ions from 0.05 to 50 nM with a LOD of 1.01 pM.

Simultaneous optical detection of mercury and lead ions was conducted using gold nanoparticles, which were modified with the DNA strand rich in guanine moieties for lead ion detection and DNA sequence abundant with thymine residues for mercury detection [36]. In the absence of metal cations, the DNA strands were hybridized with complementary ssDNA probes labeled with fluorophore Texas Red and Cy 5.5. The fluorescence signal was quenched because of the presence of gold nanoparticles in close proximity to labels. Then, the addition of metal cations led to the formation of a G-quadruplex structure (for lead ions) and a hairpin structure in the case of mercury ions. The disruption of a duplex complex enabled the recovery of the fluorescence. The proposed system allowed for lead and mercury ion detection in the range from 10 pM to 1 µM with a lower limit of detection of 51 pM for mercury and 27 pM for lead ions. There was hardly any change in the fluorescence after addition of the interfering ions such as Co^2+^, Ni^2+^, Zn^2+^, Cd^2+^, Mn^2+^, Mg^2+^, Cu^2+^, Ca^2+^, Na^+^, and K^+^, which proved the utility of the developed biosensor. The studies were also conducted with the use of serum samples, for which similar standard curves were obtained as in buffer solution with LOD for Hg^2+^ and Pb^2+^ of 121 and 128 pM, respectively.

Simultaneous detection of mercury and lead ions was also conducted using fluorescence resonance energy transfer (FRET) [35]. For that purpose, thrombin binding aptamer was applied, which was modified with donor carboxyfluorescein (FAM) and quencher 4-([4-(dimethylamino)phenyl]azo)benzoic acid (DABCYL) at 5′ and 3′ end, respectively. In the absence of metal cation, the aptamer strand was in random coiled conformation, which leads to a long distance between donor and quencher, allowing for pronounced fluorescence. After the addition of mercury or lead ions, the aptamer structure changed into a hairpin loop and G-quadruplex, respectively. A greater fluorescence decrease was observed in the case of lead ions. The linear response was recorded in the range of 0.5–30 nM and from 10 to 200 nM for lead and mercury ions, respectively. High concentrations above 1 µM of metal cations Cu^2+^, Co^2+^, Ni^2+^, Cd^2+^, Cr^3+^, Al^3+^, and Fe^3+^ caused the decrease in fluorescence, but this could be limited by the addition of phytic acid, CN- or a random DNA strand.

A fluorescent aptamer-based biosensor was applied for the detection of three metal cations: mercury, lead, and silver (Figure 30) [113]. Three different aptamers specific to each metal cation were labeled with different fluorophores: mercury ions sequence with 6-FAM, lead ions sequence with TAMRA, and silver ions sequence with Cy5. Then, the graphene oxide was introduced, which enabled the Π- Π interaction with DNA sequences. The close proximity of the fluorophore and graphene oxide serving as a quencher caused a decrease in fluorescence. This could be regained after the addition of metal cation, which resulted in the formation of a complex and release of a strand from graphene oxide structure that enabled the increase in fluorescence. The linear range of response was as follows: for Hg^2+^ ions from 0.3–14 nM, for Pb^2+^ ions 0.8–38 nM and for Ag^+^ from 4.2–210 nM. The limits of detection were 0.2, 0.5, and 2 nM for Hg^2+^, Pb^2+^, and Ag^+^, respectively.

As was mentioned above, a dual electrochemical detection of metal cations was also proposed. A good example of such is the assay developed by Qian et al. [114]. In their studies, a paper-based microfluidic chip was elaborated with a three-electrode system patterned through screen–printing. After the preparation of a carbon working electrode, gold nanoparticles were synthesized and attached to the surface, and then the cadmium and lead ions complementary strands were immobilized on the surface along with 6-mercapto-1-hexanol blocking agent. Next, the cadmium and lead ions aptamer strands were modified with ferrocene and methylene blue labels, which allowed for electrochemical detection. After the addition of the metal cations, the double-stranded structure was disrupted, and the complexes between the adjacent metal cation and aptamer strand were formed, which resulted in the decrease in the current signal. The dual detection was possible as ferrocene exhibited peak current at 0.45 V and methylene blue at −0.3 V. The peak current change for both analytes correlated with the increase in concentration in the range from 0.1 nM to 1000 nM with a LOD for cadmium ions of 23.31 pM and for lead ions of 46.23 pM. A minor signal change was recorded for interfering metal cations, including Zn^2+^, Mn^2+^, Cu^2+^, Fe^2+^, Hg^2+^, and Co^2+^. The proposed sensor was also applied to detect cadmium and lead ions in vegetable and fruit samples.

Similar to the above example, a dual detection of lead and cadmium ions was performed with the use of gold disk electrodes [115]. For that purpose, the transducer surface was firstly modified with a complementary strand for cadmium and lead aptamer and then with 6-mercapto-1-hexanol as a blocking agent. Next, the surface was modified with aptamer sequences labeled with ferrocene (for Cd^2+^ ions) or methylene blue (for Pb^2+^ ions), which differed in the values of peak potentials. In the absence of metal cations, pronounced current responses were recorded; however, after the addition of lead and cadmium ions, the electrochemical signal using the square-wave voltammetry technique decreased. The elaborated system enabled the detection of both cations in the range from 0.1 to 1000 nM with a LOD of 16.44 pM and 89.31 pM for cadmium and lead ions. The current response did not change significantly after incubation with interfering ions such as Co^2+^, Zn^2+^, Mn^2+^, Ca^2+^, Cu^2+^, and Fe^2+^.

In addition, simultaneous electrochemical detection of mercury and lead ions was realized [116]. For that purpose, the gold disk electrode was modified with lead ion aptamer and mercury ion aptamer labeled with Cu^2+^-Melamine and Nile blue molecules, respectively, as a source of the electrochemical signal. The electrode was further modified with 6-mercapto-1-hexanol to eliminate nonspecific interactions, and the proposed sensor allowed for single or dual detection. In the presence of lead, there was a conformation change of the aptamer strand, which led to the increase in the distance of the label Cu^2+^-Melamine from the electrode surface and a decrease in the current signal. Similar behavior was observed after the addition of mercury ions and the decrease in the peak current of Nile blue at −0.38 V. The developed system enabled mercury and lead ion detection in the range from 0.1–100 nM and 1 pM to 2 nM, respectively, with a LOD of 0.98 pM for lead ions and 19 pM for mercury ions. The signal change was minor after the addition of an excess of interfering cations, which confirmed the utility of the biosensor. Finally, the aptasensor was applied for the analysis of water samples. 

## 6. Microfluidic Aptamer-Based Sensors for Ions Detection

The progress in microelectronics, digitalization as well as the possibility of increasing the analysis throughput, together with the reduction in the end-user involvement, power the development of microfluidic devices (dedicated to sample preparation) integrated with optical or electrochemical aptasensors as detectors. Recently, significant attention has been paid to the development of portable and highly versatile devices dedicated to complex sample analysis outside the laboratories. Handheld devices enable a quick, precise, and mobile analysis of chosen samples such as environmental, industrial, or medical. In comparison to traditional laboratory procedures, they offer significantly reduced sample volume and shorter detection time, improved sensitivity due to high surface-to-volume ratio (also surface modified with carefully chosen nanomaterials), high throughput by parallel operation, portability, and disposability and what is no less important also real-time detection and an automated measurement process [117]. In the case of microfluidic devices integrated with biosensors and dedicated to metal ions detection, such an approach is still in its very initial stage; however, progress made for medical (biomarkers) application can be fully utilized. One such example is the device proposed by Huang, W.-H. et al. [118]. The authors proposed a device capable of simultaneous detection of Hg^2+^ and Pb^2+^ ions (Figure 31). The detection was conducted by an optical approach using graphene oxide (GO) as a quenching agent and a labeled aptamer (with FAM and HEX fluorescent dyes) solution as a reagent. Labeled aptamer sequences were mixed with 500 ppm GO solution before injection into one inlet of the microchannel, and the heavy metal sample solution was injected into another inlet. As the aptamer molecules undergo fluorescence resonance energy transfer (FRET) adsorbed at the GO surface (without metal ion in the sample), the presence of Hg^2+^ and Pb^2+^ ions were detected by measuring the change in the fluorescence intensity of the GO/aptamer suspension in relation to lead or mercury ions concentration in the sample. The authors proved high selectivity of their devices toward the heavy metal ions mix as well as interesting values of analytical parameters: linear range of 10–250 nM (i.e., 2.0–50 ppb) for Hg^2+^ ions and 10–100 nM (i.e., 2.1–20.7 ppb) for Pb^2+^ ions; the limit of detection of 0.70 ppb and 0.53 ppb for Hg^2+^ and Pb^2+^, respectively. The developed device showed a significantly lower detection limit than required by the World Health Organization (WHO) for Hg^2+^ and Pb^2+^ in drinking water (6 ppb and 10 ppb).

In the case of mercury ions detection, also an attempt was made to show the feasibility of microflow device application in its detection with the use of SERS spectroscopy in a microdroplet channel [119] (Figure 32). In this case, the aptamer-modified Au/Ag core–shell nanoparticles were used as functional sensing probes. All detection processes for the reaction between the mercury(II) ions and aptamer-modified nanoparticles were performed on a microdroplet channel. Small water droplets that included sample reagents were separated from each other by an oil phase that continuously flowed along the channel. This prevented the adsorption of aggregated colloids to the channel walls due to localized reagents within encapsulated droplets. This results in reduced residence time distributions. The limit of detection (LOD) of mercury(II) ions in water was determined by the SERS-based microdroplet sensor to be below 10 pM.

Koki Yoshida et al. used the microflow system in order to develop a DNA aptamer-linked hydrogel biochemical sensor, which allowed for repeatable detection of Ag^+^ ions [120] (Figure 33). This is an interesting and uncommon approach to developing a device based on biological molecules’ interaction with its analyte. As the molecular receptor, the appropriate aptamer sequence was used (Figure 33a). After its interaction with Ag^+^, its conformation changed. As the aptamer was embedded in hydrogel (Figure 33b), whose shape also depends on the shape of the aptamer, the overall dimensions of such hydrogel changed after its subjection to the silver ions sample (Figure 33b). The Authors also proved that the Ag^+^ ions previously bound by aptamer could be dissociated by heating and flushing through the integrated microfluidic heating device. According to the Authors, the developed microfluidic device allowed for the detection of a wide range of Ag^+^ concentrations (10^−5^–10 mM), which also include a toxic range for various aquatic organisms. It was also shown that the developed biochemical sensor could be used for long-term monitoring with high stability in ambient temperature and low power consumption [120].

The demand for microflow devices dedicated to automatic detection and quantification of chosen ions is undeniable, and the best example of such is the one developed by Chin-Chung Tseng et al. [121]. This is a typical Point-of-Care device. The Authors proposed the application of a microfluidic AuNP/aptamer sensor for monitoring the whole blood K^+^ concentration of patients with chronic kidney disease (CKD) in the point-of-care (POC) mode (Figure 34). Its working principle was based on competition in the interaction of the gold nanoparticles and potassium ions with nucleic acids aptamers. In the case of low K^+^ concentration, the aptamers secure the AuNP from aggregation and change the color of the paper strip. After increasing potassium ion concentration, the aptamers formed a rigid G-quadruplex structure and released AuNPs, which then aggregated (under the effects of NaCl reagent), and the same, changed the detection zone color for optical readout (by developed RGB analysis app software). The quantity of AuNP released from the aptasensor was proportional to the K^+^ ion concentration in the sample. The developed device was validated for serum samples after separation of whole blood and showed interesting analytical parameters: linear response in the range of 0.05–9 mM (R^2^ = 0.994), LOD of 0.01 mM, excellent agreement between obtained results versus reference ion-selective electrode method for 137 real-world CKD whole blood and 287 serum samples (R^2^ = 0.968 and R^2^ = 0.980). According to the Authors, the developed microfluidic device was easy to produce and can be used as a low-cost home self-monitoring device with a price of less than US$ 0.50 per microchip (Figure 34a). 

In another approach, cadmium ions were detected using an interdigitated electrode modified with an aptamer probe [122]. The whole detection procedure was incorporated into a microfluidic system that contained an AC electrothermal element that enhanced the transportation of cadmium ions towards IDE modified with aptamer. The proposed system allowed for cadmium ions detection in the range from 0.45 fM to 4.5 pM with a lower limit of detection of 253.16 aM. The capacitance response was negligible when interfering metal cations at a concentration of 0.45 pM were analyzed. The proposed flow system was also utilized for the experiments of spiked water and oil samples as well as a rice leaching solution.

## 7. Conclusions and Future Perspectives

This review focuses on the presentation of the latest achievements in the field of nucleic acid-based detection of metal ions. The paper covers the basic description of the DNA/metal ions interactions used for selective target binding, as well as depicts the advantages and limitations of the application of such molecules as receptors in sensor development. It should be noted that a vast number of described biosensors contain receptor layers that also comprise various nanomaterial types, which allow for the improvement of working parameters, such as sensitivity or limit of detection. This, in turn, translates into the possibility of developing detection elements of more and more compact dimensions. As there can be seen constant progress in microelectronics, materials engineering, common digitalization, or manufacturing technologies, the inevitable progress of any detection elements is related to its application in devices, allowing for increased analysis throughput both with reduced end-user involvement. Examples of such microfluidic devices integrated with optical or electrochemical nucleic acid-based detectors are also described in the presented review. Except for the above advantages, such an approach also offers a significant minimization of chemicals and sample consumption, together with a further sensitivity increasement. It can also be expected that the number of such devices’ literature examples will constantly increase, together with further development of respective material sciences and with the identification of new and more selective aptamer sequences. However, most of the innovations presented in the literature are still in the proof-of-concept state. Nonetheless, it can be expected that one of the first commercially available (bio)tests dedicated to metal ions detection will be based rather on the paper substrate and will work similarly to lateral flow assays [123]. Because of its intrinsic drawbacks, such as high detection limits or low sensitivity, such an assay should be further intensively developed to offer a truly useful solution. That is why constant development of more precise and accurate sensors is crucial, also together with devices allowing for fully automated analysis. This is strictly related, e.g., to the application of new nanomaterials in their construction for signal intensity and selectivity increasement. This review attempted to show the recent achievements in such an approach to metal ions, not only detection but also, what is crucial in most real-life applications, their precise concentration determination.

## Figures and Tables

**Figure 1 molecules-27-07481-f001:**
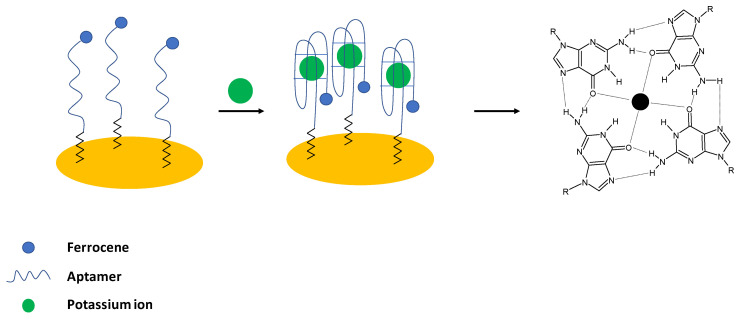
The sensing principle scheme of the electrochemical aptasensor with ferrocene labeled DNA as a receptor probe.

**Figure 2 molecules-27-07481-f002:**
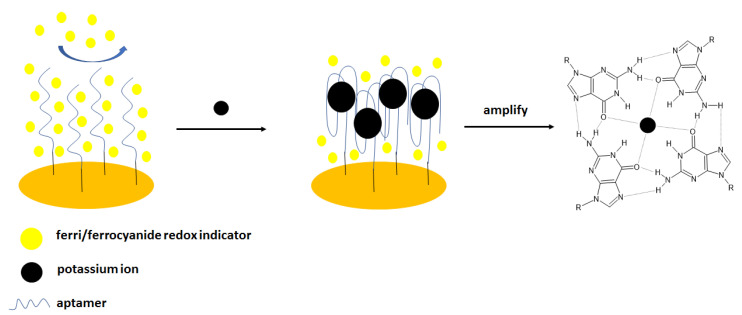
The sensing principle scheme of the impedimetric aptasensor with ferri/ferrocyanide redox couple.

**Figure 3 molecules-27-07481-f003:**
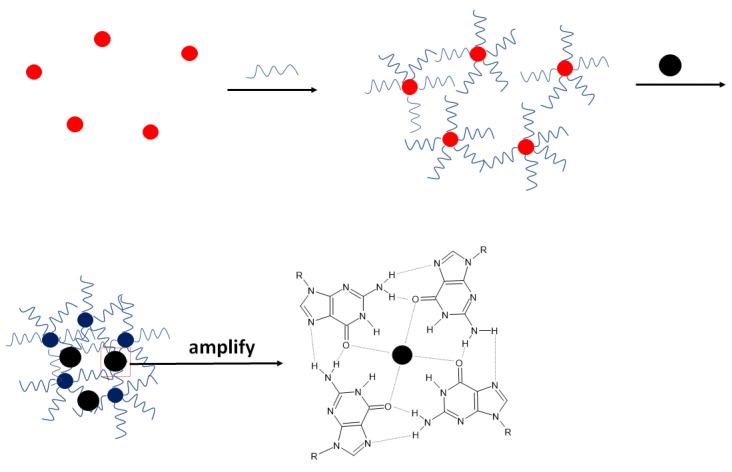
The sensing principle scheme of the optical aptasensor for potassium ion with gold nanoparticles with the use of their aggregation phenomena.

**Figure 4 molecules-27-07481-f004:**
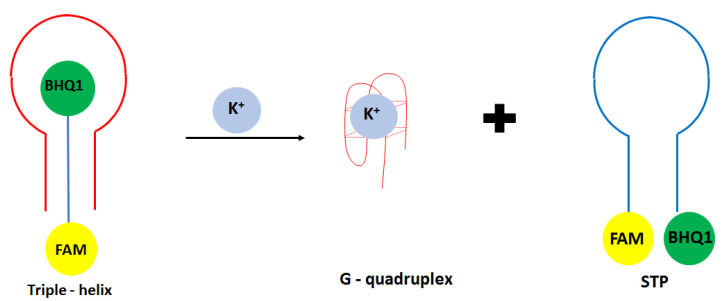
The sensing principle scheme of the fluorescence aptasensor for potassium ion with the use of triple—helix structure.

**Figure 5 molecules-27-07481-f005:**
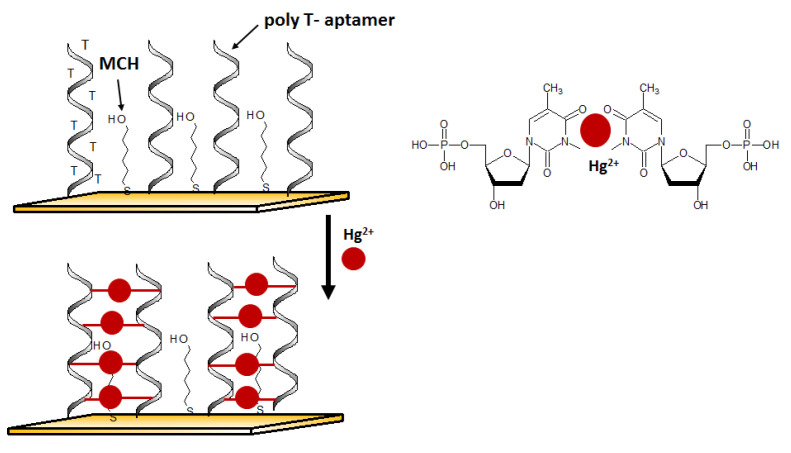
The sensing principle scheme of the electrochemical DNA sensor for mercury(II) ion.

**Figure 6 molecules-27-07481-f006:**
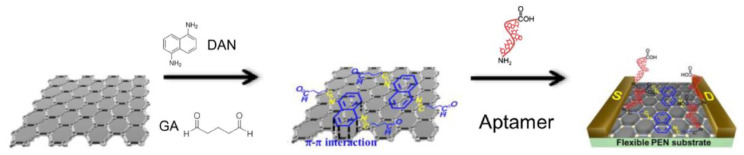
The mercury(II) ion sensing principle scheme with the use of liquid-ion gated field effect transistor (FET) transducer. Adapted with permission from An et al. [59] (Copyright © (2013) ACS).

**Figure 7 molecules-27-07481-f007:**
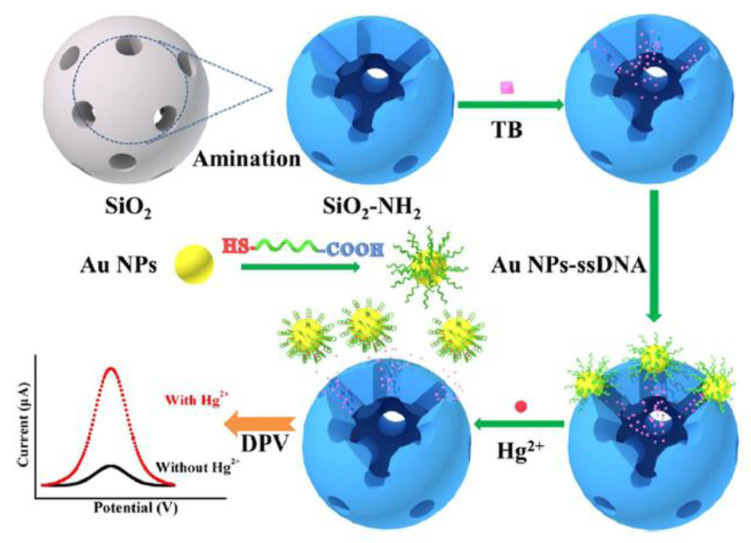
Schematic representation of the amperometric sensor for mercury ion based on analyte-controlled release of redox marker. Adapted with permission from Ma et al. [60] (Copyright © (2020) ACS).

**Figure 8 molecules-27-07481-f008:**
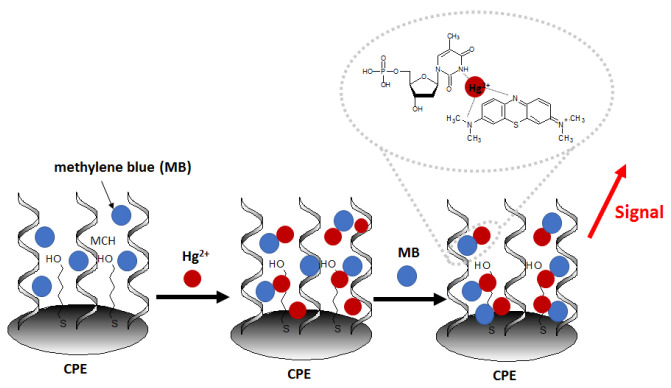
Schematic representation of construction and working principle of aptasensor using electrospun nanofibers polyethersulfone and quantum dots on carbon paste electrode.

**Figure 9 molecules-27-07481-f009:**
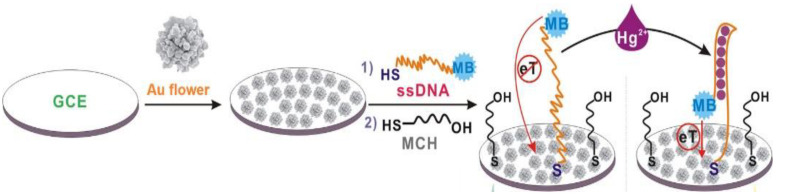
Schematic illustration of signal-on switch aptasensor using “Au nanoflowers”-decorated electrode. Adapted with permission from Zhang et al. [63] (Copyright © (2018) ACS).

**Figure 10 molecules-27-07481-f010:**
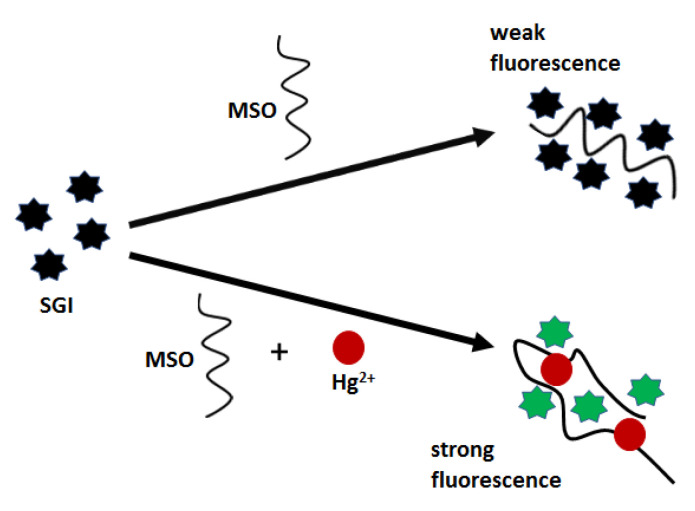
Schematic diagram of the “turn-on” biosensor for spectrofluorometric detection of Hg^2+^.

**Figure 11 molecules-27-07481-f011:**
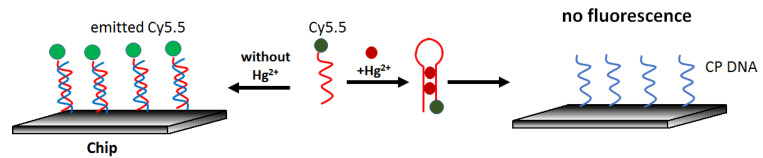
Schematic diagram of the waveguide-based aptasensor for the determination of Hg^2+^ by means of fluorescence turn-off due to the blocking hybridization of fluorescent probe by mercury ions.

**Figure 12 molecules-27-07481-f012:**
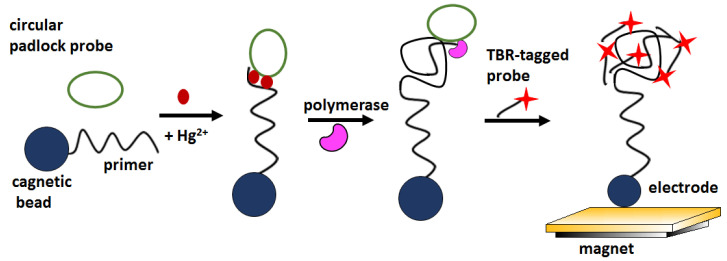
Schematic illustration of detecting mercury ions via ECL aptasensor and RCA amplification method.

**Figure 13 molecules-27-07481-f013:**
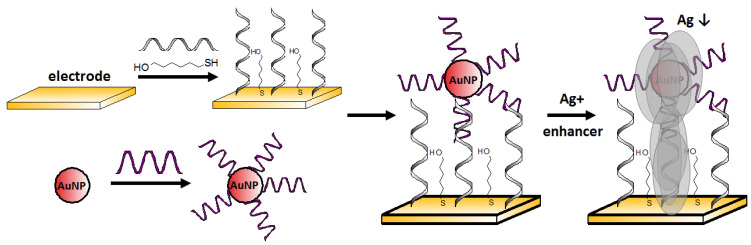
Electrochemical biosensor fabrication procedure and silver ion detection in the sandwich mode.

**Figure 14 molecules-27-07481-f014:**
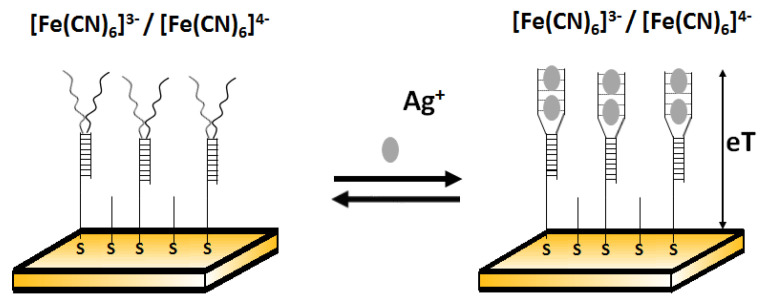
Electrochemical biosensor fabrication with Y-shaped dsDNA in the receptor layer before (**left**) and after (**right**) adding silver ions.

**Figure 15 molecules-27-07481-f015:**
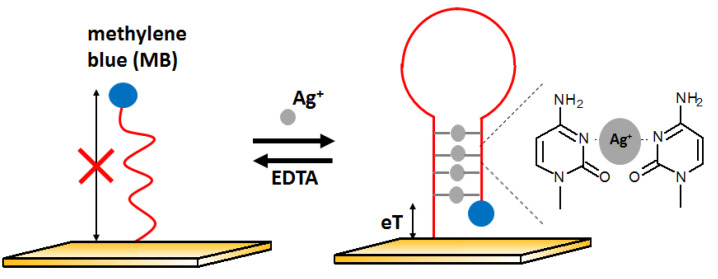
Oligonucleotide-based electrochemical biosensor in the receptor layer before (**left**) and after (**right**) adding silver ions.

**Figure 16 molecules-27-07481-f016:**
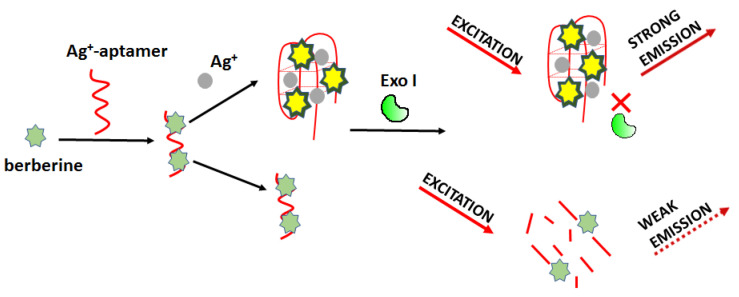
Schematic representation of Ag^+^ detection based on Exonuclease I-aided fluorescence aptasensor.

**Figure 17 molecules-27-07481-f017:**
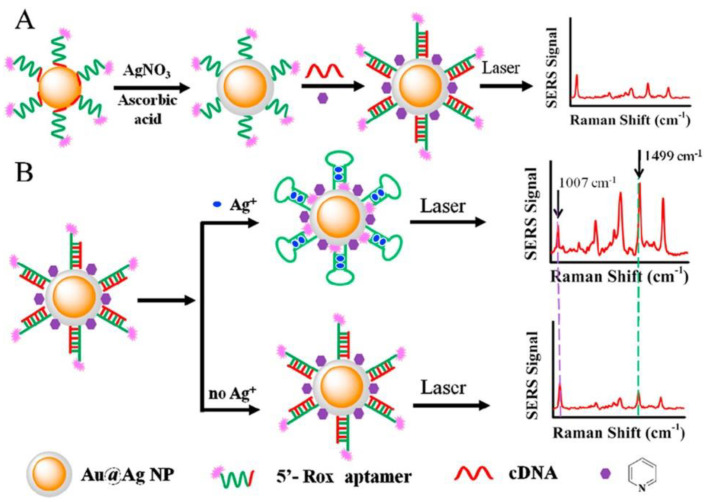
Schematic representation of Ag^+^ detection with ratiometric SERS aptasensor. (**A**) nanoparticles Ag core-shell preparation; (**B**) Ag ion detection. Adapted with permission from Wu et al. [76] (Copyright © (2018) Elsevier).

**Figure 18 molecules-27-07481-f018:**
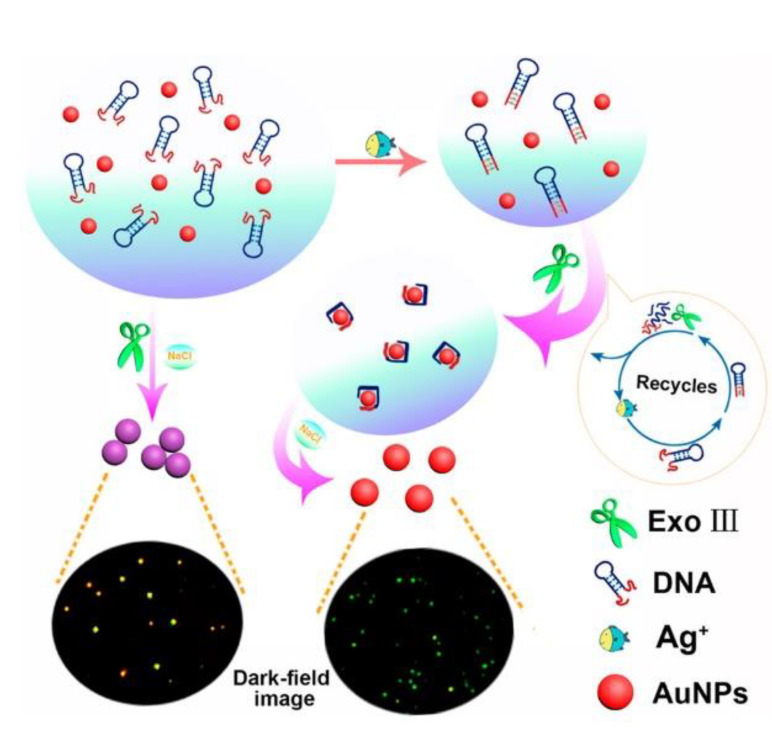
Scheme of DNA based Ag^+^ sensing through Dark-Field Microscopy. Adapted with permission from Li et al. [80] (Copyright © (2018) ACS).

**Figure 19 molecules-27-07481-f019:**
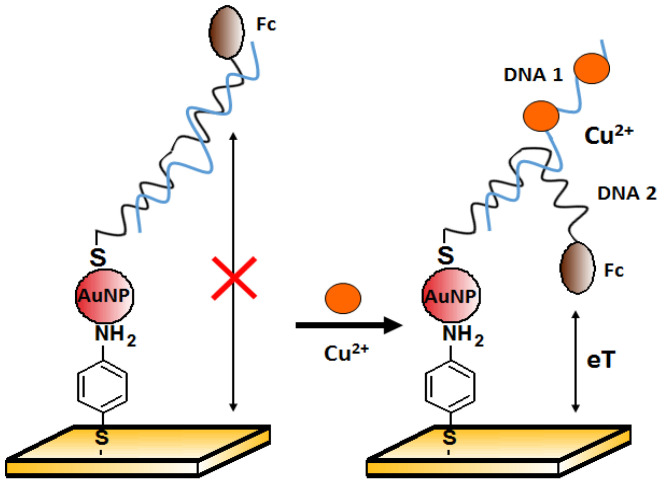
Scheme of the electrochemical sensor with AuNPs for Cu^2+^ ion detection.

**Figure 20 molecules-27-07481-f020:**
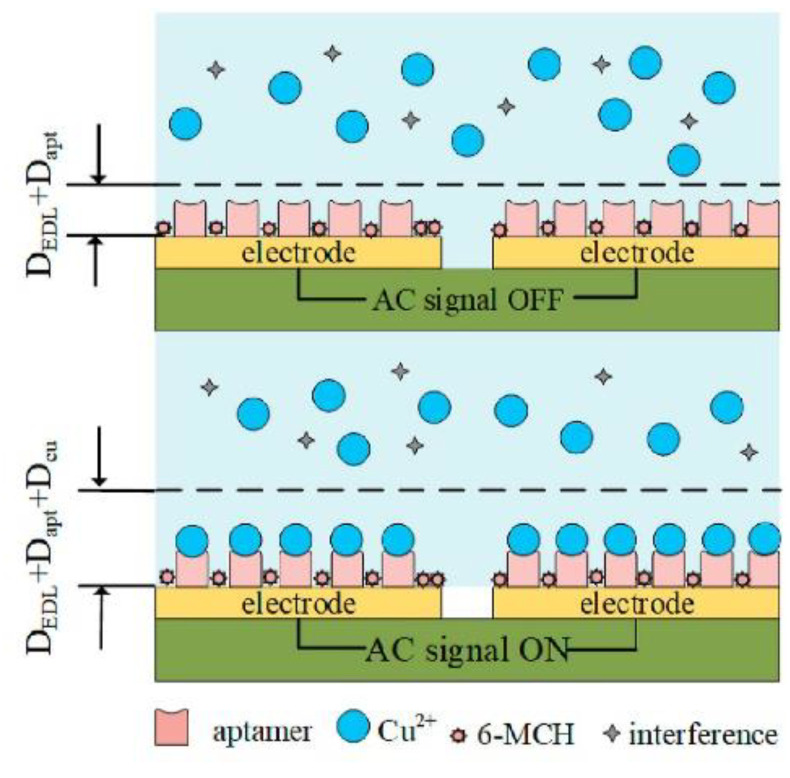
Scheme of the copper ions electrochemical sensor with the gold-plated coplanar electrode array. Adapted with permission from Qi et al. [86] (Copyright © (2021) Elsevier).

**Figure 21 molecules-27-07481-f021:**
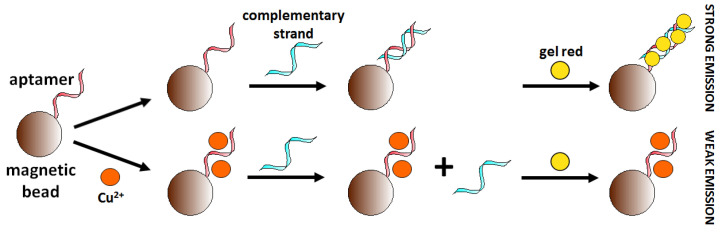
Scheme of the fluorescent aptasensor for copper ions.

**Figure 22 molecules-27-07481-f022:**
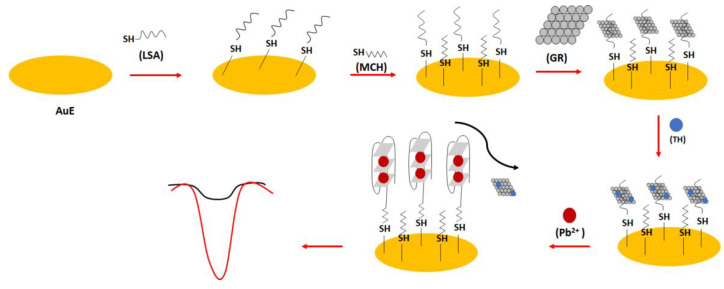
The sensing principle scheme of the graphene/thionine based electrochemical aptasensor towards lead ions.

**Figure 23 molecules-27-07481-f023:**
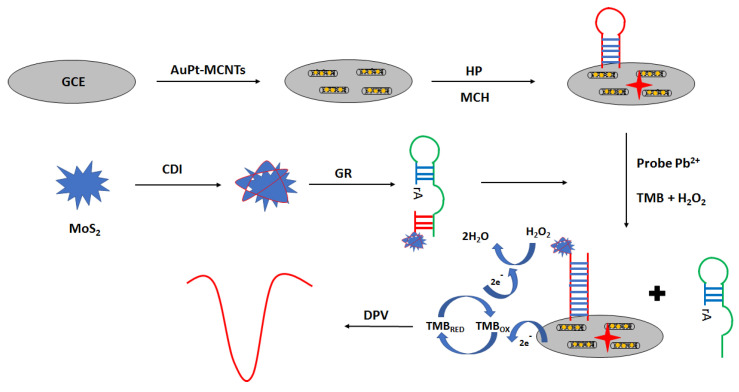
The sensing principle scheme of the electrochemical aptasensor towards lead ions with the application of peroxidase-like GR sequence aptamer—functionalized 3D—flower MoS_2_ microsphere hybrid as signaling probes.

**Figure 24 molecules-27-07481-f024:**
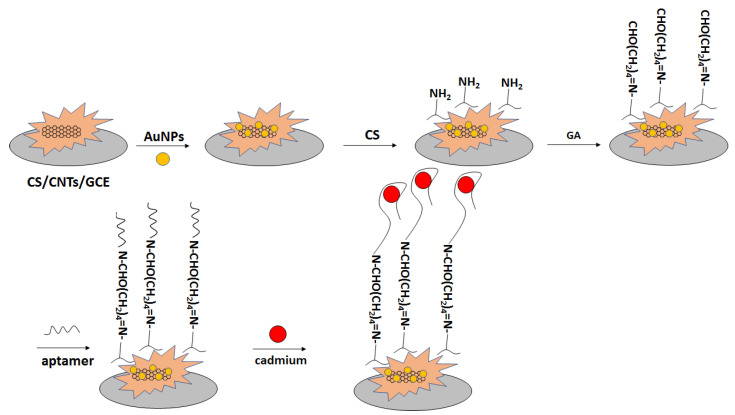
The sensing principle scheme of the electrochemical aptasensor toward cadmium ions with the application of a glassy carbon electrode modified with carbon nanotubes in chitosan solution and impedance spectroscopy in the presence of a ferri/ferrocyanide redox indicator as an analytical technique.

**Figure 25 molecules-27-07481-f025:**
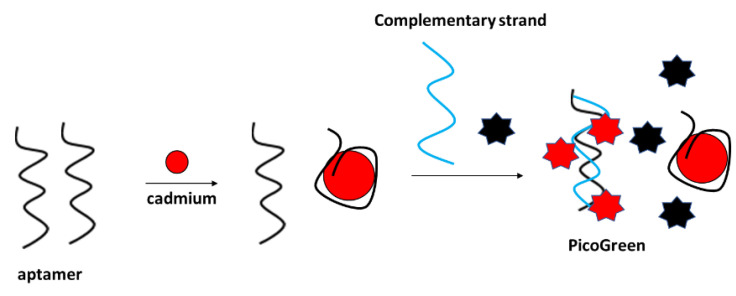
The sensing principle scheme of the fluorescence aptasensor towards cadmium ion based on PicoGreen indicator selectively interacting with double-stranded DNA.

**Figure 26 molecules-27-07481-f026:**
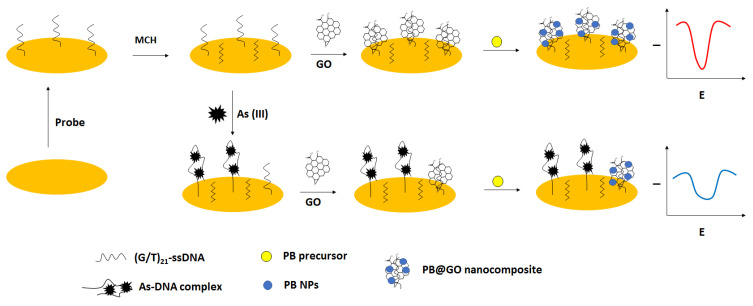
The sensing principle scheme of the electrochemical aptasensor towards arsenic ion based on in situ generation of Prussian blue nanoparticles.

**Figure 27 molecules-27-07481-f027:**
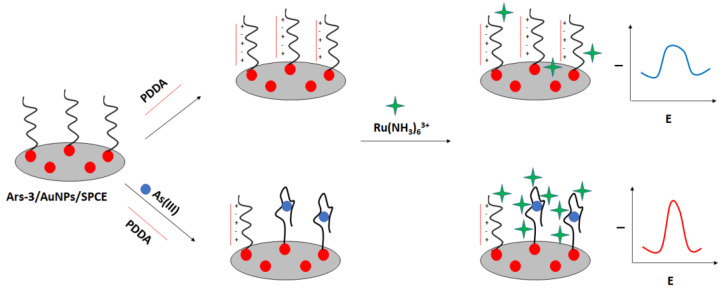
The sensing principle scheme of the label-free signal on electrochemical sensor towards arsenic ions.

**Figure 28 molecules-27-07481-f028:**
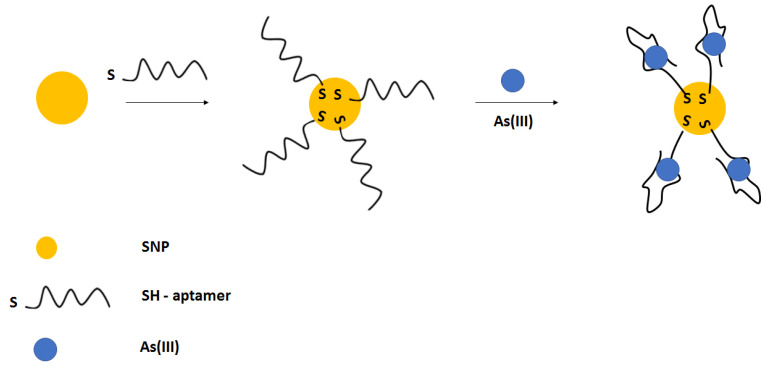
The sensing principle scheme of the colorimetric assay for arsenic ions detection.

**Figure 29 molecules-27-07481-f029:**
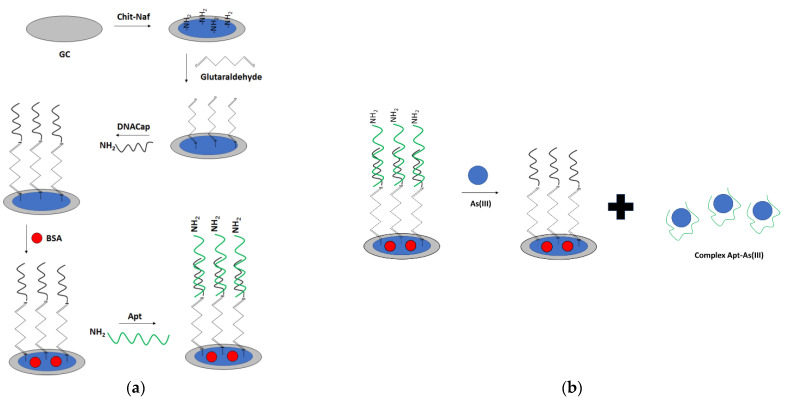
The sensing principle scheme of a dual optical and electrochemical sensor towards arsenic ion. (**a**) sensing layer preparation; (**b**) arsenic ion detection.

**Figure 30 molecules-27-07481-f030:**
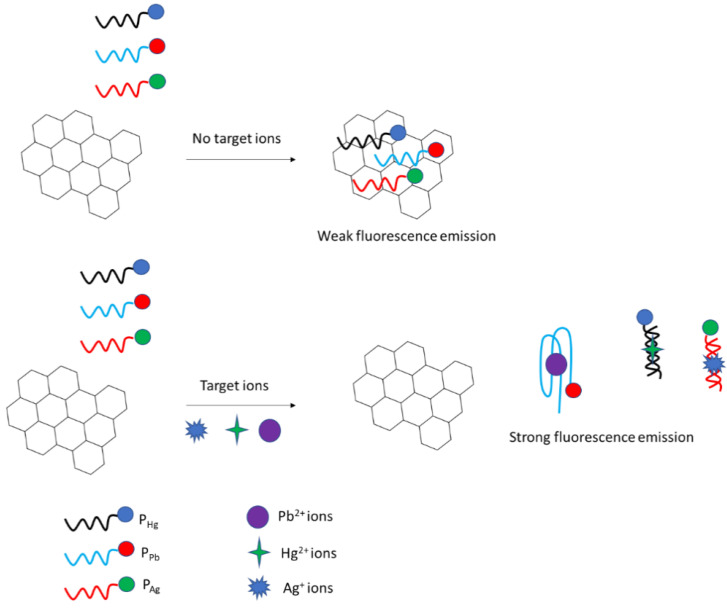
A fluorescent biosensor for detection of lead, mercury and silver ions.

**Figure 31 molecules-27-07481-f031:**
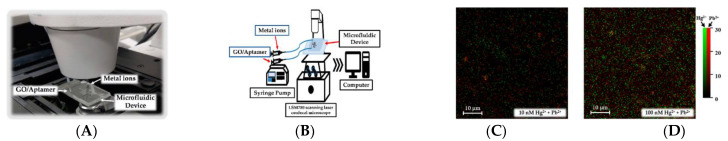
(**A**) experimental setup; (**B**) Schematic integration of experimental setup; (**C**,**D**) exemplary result of simultaneous detection of Hg^2+^ and Pb^2+^ ions in two concentrations, 10 nM and 100 nM respectively [118].

**Figure 32 molecules-27-07481-f032:**
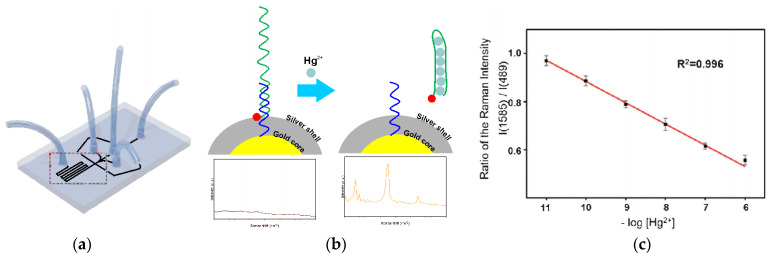
(**a**) Microdroplet channel layout for the SERS detection of mercury(II) ions; (**b**) SERS signal change for the microdroplets measured according to aptamer DNA release from the nanoparticle surface in the presence of mercury(II) ions, forming a stable T-Hg^2+^-T mediated hairpin structure; (**c**) Peak intensity ratios (1585 (Cy3)/489(PDMS)) as a function of mercury(II) concentration [119].

**Figure 33 molecules-27-07481-f033:**
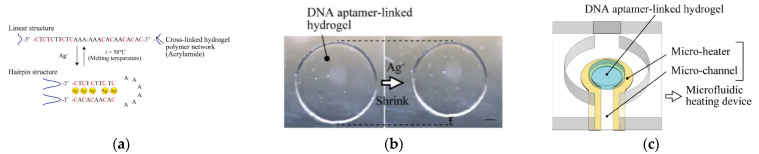
(**a**) The structure of Ag-DNA aptamer reversibly changes through binding and dissociating the silver ions (**b**) The volume change of the DNA aptamer-linked hydrogel as the sensor response toward 10 mM Ag^+^ ions. The scale bar is 500 μm (**c**) Scheme of the DNA aptamer-linked hydrogel sensor integrated with the microfluidic heating device for repeatable detection of chemical substances [120].

**Figure 34 molecules-27-07481-f034:**
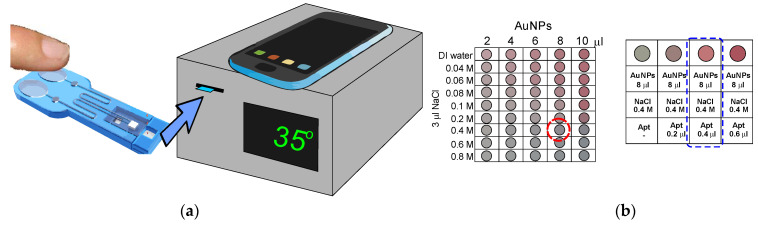
(**a**) The example of aptamer biochip to K^+^ detection and the dedicated device for operation, (**b**) Reaction color for different volumes of AuNPs and concentrations of 3 mLNaCl together with reaction color for 8 mL AuNPs, 3 mL NaCl (0.4 M) and different volumes of aptamers [121].

**Table 1 molecules-27-07481-t001:** Comparison of aptamers.

DNA	RNA	Peptide
oligonucleotide	oligonucleotide	small peptide with a single variable loop region that binds target
secondary and tertiary structures	secondary and more diverse tertiary structures	3D structures constrained by scaffolds—reduced flexibility
less reactive and more stable than RNA aptamers due C-H bond at 2′ position of deoxyribose	susceptible to nucleophilic attacks, less stable than DNA aptamers due to the presence of a reactive hydroxyl group at 2′ position of the ribose moiety	available in large quantities through chemical synthesis or bacterial expression
cheap and easy to produce inin vitroSELEX process	requires reverse transcription in SELEX process	produced and selectedin vivothrough yeast two hybrid or biopanning

## Data Availability

Not applicable.

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
