# Peer review of "Recent Achievements in Electrochemical and Optical Nucleic Acids Based Detection of Metal Ions"

_molecules, 2022, doi:10.3390/molecules27217481_

Round 1
Reviewer 1 Report
This review provides current progress in nucleic acids application in monitoring environmentally and clinically important metal ions in the electrochemical or optical manner. However, there are still many problems for practical application. Thus, the manuscript should be made minor revision before publication in Molecules. Several issues to be addressed are as follows:
1. The article is about the detection of metal ions based on aptamers. So the article title would be more appropriately changed to “Recent achievements in electrochemical and optical aptamers based detection of metal ions”.
2. The layout of the images in the article is not very aesthetically pleasing. We suggest to re-arrange the images.
Author Response
This review provides current progress in nucleic acids application in monitoring environmentally and clinically important metal ions in the electrochemical or optical manner. However, there are still many problems for practical application. Thus, the manuscript should be made minor revision before publication in Molecules. Several issues to be addressed are as follows:
- The article is about the detection of metal ions based on aptamers. So the article title would be more appropriately changed to “Recent achievements in electrochemical and optical aptamers based detection of metal ions”.
Thank You very much for this comment. However we believe that title “Recent achievements in electrochemical and optical nucleic acids based detection of metal ions” better describes the manuscript content, also in the point of view of ambiguous aptamers definition. Aptamers are often defined as oligonucleotides that binds to certain analyte. However according to its initial definition coined by Andy Ellington and stemed from the Latin term “aptus” which mean to fit, and “meros” meaning part, aptamers should also appropriately adjust its spatial structure to the respective analyte. In the case of metal ions detection often only some interaction with only certain atoms in oligonucleotide structure is used, e.g. nitrogen during mercury ion detection. It not necessary should lead to the change in the oligonucleotide structure and to the fitting of its structure to the detecting analyte. That is why, in order to avoid some inconsistencies, we choose the title “Recent achievements in electrochemical and optical nucleic acids based detection of metal ions”.
- The layout of the images in the article is not very aesthetically pleasing. We suggest to re-arrange the images.
The idea of the article was to show basic principles of oligonucleotide interaction with metal ion and detection of such interactions. However some images were changed or rearranged.
Reviewer 2 Report
This manuscript summarized the recent development of the nucleic acids based sensors for metal ions, it is attracting to the readers, I suggest publishing this paper after being minor revised.
1. The quality of several figures, such as Figure 4, 5, 6, ect., is too poor.
2. The English needs to be improved.
Author Response
This manuscript summarized the recent development of the nucleic acids based sensors for metal ions, it is attracting to the readers, I suggest publishing this paper after being minor revised.
1 The quality of several figures, such as Figure 4, 5, 6, ect., is too poor.
The idea of the article was to show basic principles of oligonucleotide interaction with metal ion and detection of such interactions. However some images were changed or rearranged.
2 The English needs to be improved.
Authors carefully reread the article and corrected founded mistakes.
Reviewer 3 Report
Manuscript ID: Molecules-1952808
Herein, the authors of the article have given a complete report on electrochemical and optical methods based on aptamers. In this context, it is better to report the following in this review. The article can be accepted after the following issues are reported.
1. There is no mention of table number two in the text.
2. Regarding FRET methods, it is better for the authors to fully explain this method first and explain the conditions of this method such as the distance between the quencher and the emitter, the choice of aptamer type and the type of sequences.
3. In some cases, it is mentioned in the paper that ions are detected by aptamers through G-quadruplex formation. It is better for the authors to fully explain the conditions for the formation of this type of structure.
4. Regarding FET methods, it is necessary for the authors to explain the principles of this method.
5. The authors present an analysis of optical and electrochemical aspects to explain the sensitivity, selectivity, or usability of these methods in complex environments such as plasma or wastewater.
6. The authors should also report smartphone-based detection of metal ions.
7. Authors should refer to the following articles:
2022, Anal. Chem. 94, 22, 8005
2021, Sensors and Actuators B: Chemical 348, 130658
Author Response
.
